# Efficient Adversarial Attacks on High-dimensional Offline Bandits

**Seyed Mohammad Hadi Hosseini, Amir Najafi & Mahdieh Soleymani Baghshah**
Department of Computer Engineering
Sharif University of Technology
{hadi.hosseini17,amir.najafi,soleymani}@sharif.edu

## Abstract

Bandit algorithms have recently emerged as a powerful tool for evaluating machine learning models, including generative image models and large language models, by efficiently identifying top-performing candidates without exhaustive comparisons. These methods typically rely on a reward model—often distributed with public weights on platforms such as Hugging Face—to provide feedback to the bandit. While online evaluation is expensive and requires repeated trials, offline evaluation with logged data has become an attractive alternative. However, the adversarial robustness of offline bandit evaluation remains largely unexplored, particularly when an attacker perturbs the reward model (rather than the training data) prior to bandit training. In this work, we fill this gap by investigating, both theoretically and empirically, the vulnerability of offline bandit training to adversarial manipulations of the reward model. We introduce a novel threat model in which an attacker exploits offline data in high-dimensional settings to hijack the bandit's behavior. Starting with linear reward functions and extending to nonlinear models such as ReLU neural networks, we study attacks on two Hugging Face evaluators used for generative model assessment: one measuring aesthetic quality and the other assessing compositional alignment. Our results show that even small, imperceptible perturbations to the reward model's weights can drastically alter the bandit's behavior. From a theoretical perspective, we prove a striking high-dimensional effect: as input dimensionality increases, the perturbation norm required for a successful attack decreases, making modern applications such as image evaluation especially vulnerable. Extensive experiments confirm that naive random perturbations are ineffective, whereas carefully targeted perturbations achieve near-perfect attack success rates. To address computational challenges, we design efficient heuristics that preserve almost $100\%$ success while dramatically reducing attack cost. In parallel, we propose a practical defense mechanism that partially mitigates such attacks, paving the way for safer offline bandit evaluation. Finally, we validate our findings on the UCB bandit and provide theoretical evidence that adversaries can delay optimal arm selection proportionally to the input dimension. Code is publicly available at: `https://github.com/hadi-hosseini/adversarial-attacks-offline-bandits`.

## 1 Introduction

The multi-armed bandit (MAB) framework is a cornerstone of online learning, balancing exploration and exploitation to efficiently identify the optimal choice (or "arm") within a limited time horizon. MAB algorithms have been widely applied in domains such as online recommendation systems (Kveton et al., 2015; Chapelle et al., 2014; Li et al., 2010), adaptive clinical trials (Kuleshov & Precup, 2014), and dynamic pricing (Tullii et al., 2024). More recently, they have been adopted for evaluating generative models—from diffusion models (Rezaei et al., 2025; Hu et al., 2025; 2024b; Ho et al., 2020; Rombach et al., 2022) and GANs (Rezaei et al., 2025; Hu et al., 2024a; Karras et al., 2019) to large language models (LLMs) such as ChatGPT (Hu et al., 2025; Dai et al., 2025; Achiam et al., 2023), Gemini (Hu et al., 2025; Dai et al., 2025; Team et al., 2023), and DeepSeek (Hu et al., 2025; Guo et al., 2025). In this setting, bandit algorithms (e.g., Upper Confidence Bound (UCB)

from Auer (2002)) offer a sample-efficient alternative to naive pairwise comparisons by rapidly identifying the best-performing model.

While online evaluation remains effective, it is often costly and impractical for rapidly evolving generative models. As a result, practitioners increasingly turn to offline evaluation, where a fixed logged dataset is reused to compare algorithms (e.g., UCB, ETC (Garivier et al., 2016), $\varepsilon$-greedy (Dann et al., 2022)) or tune hyperparameters without repeated data collection. Despite its practicality, this setting introduces two overlooked risks. First, it is vulnerable to adversarial manipulation that could bias the selection process. Second, it is prone to overfitting to the reward model when multiple degrees of freedom are available. For instance, recent work has shown a tendency to test numerous performance metrics or continuously adjust metric hyperparameters on the same logged dataset when training MABs (Rezaei et al., 2025; Hu et al., 2024b; Saito & Nomura, 2024).

We aim to provide partial answers to the above concerns across a range of settings. Specifically, we study adversaries that manipulate only the reward model (rather than the data samples) in order to hijack the behavior of a bandit algorithm. While prior work has examined training-time adversarial attacks on bandits—typically by altering reward values or samples during online training (see Appendix A)—no prior research has considered attacks on the reward model *before* training. We fill this gap by showing that even a weak adversary can succeed in this setting when the data is high-dimensional. In our threat model, the adversary has access to the offline (logged) dataset but cannot modify the samples, nor can it interfere with the learner's training procedure; its only leverage is to perturb the reward model prior to training.

We analyze two reward-model scenarios. First, we study linear reward functions in high-dimensional spaces. This setting, though simplified, offers clarity and valuable insights into the mechanics of our attack. Our results show that even small perturbations to the weights of a linear model can dramatically alter the bandit's behavior. Second, we extend our analysis to nonlinear reward models such as deep neural networks (LeCun et al., 2015), which are widely used in practice. For instance, modern image-based generative models are often evaluated with neural reward functions such as CLIP (Radford et al., 2021; LAION-AI, 2021), BLIP (Li et al., 2022; Xu et al., 2023), and VQA models (Hu et al., 2023; Singh & Zheng, 2023; Cho et al., 2024). These models are typically open-source, with publicly available weights on platforms such as Hugging Face—making them convenient, but also vulnerable. We select two real-world image reward models from Hugging Face and apply our attack to them: one for assessing compositionality alignment and the other for evaluating aesthetic quality. We demonstrate that exposing verifier weights can be exploited: by constructing a linear approximation of the reward model, an adversary can craft perturbations of imperceptibly small norm that nonetheless yield highly effective attacks. Moreover, we show that this linear approximation becomes increasingly accurate as the network's hidden layer width grows, consistent with predictions from Neural Tangent Kernel (NTK) theory (Jacot et al., 2018; Golikov et al., 2022). We identify several key insights and introduce novel algorithms that enable highly effective attacks on offline bandit training, supported by both theoretical analysis and empirical validation. Our main contributions are:

**Theoretical foundations.** We show that effective attacks require solving a targeted optimization problem; naive random perturbations fail to compromise the reward model. Empirically, we observe an inverse relationship between dimension and attack cost: as input dimensionality increases, the required perturbation norm decreases, making high-dimensional data (e.g., images) especially vulnerable. We then complement empirical observations via theoretical support. Formally, for a $K$-armed bandit over a $d$-dimensional input space with horizon $T$, we prove that full-trajectory attacks are feasible whenever $d \geq KT$ (Theorem 3.3), and that the $\ell_2$-norm of the required perturbation shrinks as $\widetilde{\mathcal{O}}(d^{-1/2})$ (Theorem 3.4). Our analysis is based on tools such as high-dimensional concentration inequalities and random matrix theory.

**Computational efficiency.** Solving the full optimization problem to hijack an entire trajectory can be computationally expensive. We design heuristic methods that dramatically reduce this cost while maintaining near-perfect attack success rates. We validate these attacks across multiple bandit algorithms, focusing on UCB but also extending to ETC and $\epsilon$-greedy, showing that our method generalizes beyond a single algorithm.

**Synthetic and real-world evaluation.** We first validate our approach on synthetic data, then extend to real-world settings by targeting two widely used Hugging Face evaluators for generative models:

one measuring aesthetic quality and the other compositional alignment. In both cases, our attacks remain highly effective.

**Defense.** We propose a simple defense mechanism that can partially mitigate attacks under certain conditions, reducing attack success rates. However, a complete defense remains open, highlighting an important direction for future research.

## 2 PROBLEM DEFINITION AND NOTATION

For $K \in \mathbb{N}$, let $[K] = \{1, 2, \ldots, K\}$. We consider a stochastic multi-armed bandit problem with $K$ arms, where each arm $i \in [K]$ is associated with a data-generating distribution $P_i$ over $\mathbb{R}^d$. Here, $d$ denotes the dimension of the input space. At each round, when the bandit pulls arm $i$, it receives a sample $\mathrm{X} \sim P_i$, drawn independently from past rounds. The *reward* is then computed as $r(\mathrm{X})$, where $r : \mathbb{R}^d \to \mathbb{R}$ is a known reward function that may be perturbed adversarially before training begins. We study two instantiations of the reward function:

1. **Linear model:** $r(\mathrm{X}) = \mathrm{w}^\top \mathrm{X}$, with a fixed parameter vector $\mathrm{w} \in \mathbb{R}^d$.

2. **Neural network model:** $r(\mathrm{X}) = \mathrm{NN}_{\boldsymbol{\theta}}(\mathrm{X})$, where $\mathrm{NN}_{\boldsymbol{\theta}}$ is a feedforward network with parameters $\boldsymbol{\theta} \in \mathbb{R}^D$ and some fixed activation function $\sigma(\cdot) : \mathbb{R} \to \mathbb{R}$. The parameter dimension is $D = d \cdot W_1 + W_1 \cdot W_2 + \cdots + W_{L-1} \cdot W_L + W_L$, for a network with $L$ hidden layers of widths $W_1, W_2, \ldots, W_L$, respectively.

Throughout, we focus on the high-dimensional regime: either $d \gg 1$ (linear case) or $\max_i W_i \gg 1$ (neural network case). We assume arm $i^* \in [K]$ is optimal, i.e., $\mathbb{E}_{P_{i^*}}[r(\mathrm{X})] > \mathbb{E}_{P_i}[r(\mathrm{X})]$ for $i \neq i^*$, where $\mathbb{E}[\cdot]$ is expectation operator. Hence, we expect the bandit algorithm to (with high probability) learn to pull the optimal arm $i^*$ after some number $T \geq 1$ of exploratory trials across the $K$ arms. The value $T$ is referred to as the *horizon*. Suppose we are given $K$ logged datasets consisting of independent samples from each arm distribution $P_1, \ldots, P_K$. Specifically, let $\mathcal{D}_i = \{\mathrm{X}_j^{(i)} | j \in [n_i]\}$, be a dataset of size $n_i 2$ containing i.i.d. samples from $P_i$. When the bandit pulls arm $i$ for the $t_i$-th time, it receives the sample $\mathrm{X}_{t_i}^{(i)}$, for $t_i \in [n_i]$. We assume that each dataset $\mathcal{D}_{1:K}$ contains sufficiently many samples so that, throughout the $T$ rounds, the bandit always observes a fresh sample upon each arm pull.

**Our Problem:** Our goal is to design an adversarial attack on the reward function $r(\cdot)$ such that, with high probability, the bandit fails to identify and exploit the optimal arm. Assume an adversary $\mathscr{A}$ with access to the offline logged datasets $\mathcal{D}_1, \ldots, \mathcal{D}_K$, and the ability to perturb the reward model $r(\cdot)$. In the linear case, the adversary modifies the parameter $\mathrm{w}$ to $\mathrm{w} + \boldsymbol{\delta}$ for some perturbation vector $\boldsymbol{\delta} \in \mathbb{R}^d$. In the neural network case, the adversary modifies the parameter vector $\boldsymbol{\theta}$ to $\boldsymbol{\theta} + \boldsymbol{\delta}$, where $\boldsymbol{\delta} \in \mathbb{R}^D$. In both settings, the perturbation norm is chosen as small as possible while still guaranteeing a prescribed level of damage to the bandit's trajectory, for example, ensuring that the number of times the optimal arm $i^*$ is pulled falls below some threshold $T' < T$. In stronger attack scenarios, $\mathscr{A}$ may even enforce a pre-specified target trajectory for the multi-armed bandit.

## 3 METHOD

In this section, we focus on UCB (Auer, 2002) algorithm for simplicity, but our findings extend naturally to ETC and $\varepsilon$-greedy, as detailed in Appendix D. UCB begins by pulling each arm once, and then proceeds by selecting arms in a way that balances exploitation and exploration: it favors arms with higher observed average rewards, while a carefully designed exploration term encourages underexplored arms to be pulled. It is theoretically known that UCB achieves a regret bound of $\mathcal{O}(\log T)$, which is optimal up to constants, making it widely used in practice. Let $A_t = A_t(\boldsymbol{\delta})$ for $t \in [T]$ denote the arm chosen at step $t$ when the adversary $\mathscr{A}$ selects the perturbation vector $\boldsymbol{\delta}$. Then, the algorithm proceeds for $t = 1, \ldots, T$ as follows:

- For $t = 1, \ldots, K$, pull each arm once. That is, $A_t = t$.

- For $t \geq K$, select the next arm according to the following deterministic maximization:

$$A_{t+1}(\boldsymbol{\delta}) \triangleq \operatorname*{arg\,max}_{i \in [K]} \left\{ \Lambda_t(i; \boldsymbol{\delta}) \triangleq \frac{1}{N_i(t)} \sum_{j=1}^{N_i(t)} r\left( \mathrm{X}_j^{(i)}; \boldsymbol{\delta} \right) + \sqrt{\frac{2 \log t}{N_i(t)}} \right\}, \qquad (1)$$

where $N_i(t)$ denotes the number of times arm $i$ has been pulled up to step $t$, and $r(\mathrm{X}; \boldsymbol{\delta})$ denotes the adversarially perturbed reward value of input X given the perturbation vector $\boldsymbol{\delta}$, e.g., $r(\mathrm{X}; \boldsymbol{\delta}) = (\mathrm{w} + \boldsymbol{\delta})^\top \mathrm{X}$ in the linear case. Also, $\Lambda_t(i; \boldsymbol{\delta})$ is called the UCB score of arm $i$ at time step $t$.

The adversary aims to keep $\|\boldsymbol{\delta}\|_2$ as small as possible. However, depending on its objective, $\mathscr{A}$ may design $\boldsymbol{\delta}$ to achieve one of the following goals:

**Full Trajectory Attack.** In this scenario, $\mathscr{A}$ attempts to force the bandit to follow a completely predetermined, arbitrary, and fixed target trajectory $\widetilde{A}_t \in [K]$ for all $t \in [T]$. This requires solving the following constrained optimization problem:

$$\boldsymbol{\delta}^* \triangleq \operatorname*{arg\,min}_{\boldsymbol{\delta}} \|\boldsymbol{\delta}\|_2^2 \quad \text{subject to} \quad \Lambda_t(\widetilde{A}_t; \boldsymbol{\delta}) > \Lambda_t(i; \boldsymbol{\delta}), \quad \forall \big(K < t \leq T, \, i \neq \widetilde{A}_t \big). \quad (2)$$

The optimization in equation 2 requires $(T - K)(K - 1)$ constraints. In the linear reward model, these constraints are linear in $\boldsymbol{\delta}$.

**Trajectory-Free Attack.** A less restrictive attack requires the bandit to *avoid* pulling the optimal arm at a given set of time-steps $\mathcal{T} \subset [T]$. Note that $\mathcal{T}$ may include all steps from $K + 1$ to $T$. Unlike the full trajectory attack, here $\mathscr{A}$ only ensures that $i^*$ is not selected, without prescribing exactly which arm must be pulled. One simplified formalization is as follows: consider an arbitrary trajectory $\widetilde{A}_t \in [K]$ for all $t \in [T]$ such that $\widetilde{A}_t \neq i^*$. Then we can define:

$$\boldsymbol{\delta}^* \triangleq \operatorname*{arg\,min}_{\boldsymbol{\delta}} \|\boldsymbol{\delta}\|_2^2 \quad \text{subject to} \quad \Lambda_t(\widetilde{A}_t; \boldsymbol{\delta}) > \Lambda_t(i^*; \boldsymbol{\delta}), \quad \forall \, K < t \leq T. \quad (3)$$

Here, the auxiliary sequence $\widetilde{A}_t \neq i^*$ for $K < t \leq T$ is arbitrary and can be optimized (e.g., via heuristics) by the attacker to further reduce attack strength. In our experiments, we use a simple round-robin strategy for $\widetilde{A}_t$. The formulation in equation 3 requires only $T - K$ constraints, assuming $\mathcal{T}$ includes all steps from $K + 1$ to $T$. Trajectory-free attacks thus reduce the number of constraints by a factor of $K - 1$, significantly lowering computational cost when $K$ is large. However, for large $T$, they may still be expensive and difficult to design. To address this, we propose an online attack strategy.

**Online Score-Aware (OSA) Attack.** Consider a fixed set of time-steps $\mathcal{T} \subset [T]$. We iterate over $t \in \mathcal{T}$ in increasing order. Let $\boldsymbol{\delta}_{t-1}^*$ denote the optimal attack up to time step $t - 1$. At each step $t$, the attacker checks whether the UCB score $\Lambda_t(i^*; \boldsymbol{\delta}_{t-1}^*)$ is the largest among all arms. If not, we simply move on to the next $t$. Otherwise, we add the constraint

$$\Lambda_t(\widetilde{i}_t; \boldsymbol{\delta}) > \Lambda_t(i^*; \boldsymbol{\delta}),$$

where $\widetilde{i}_t$ is the index of the runner-up arm with the second highest UCB score at time $t$. The solution $\boldsymbol{\delta}_t^*$ is then updated using the additional constraint, and the process continues until the end of $\mathcal{T}$. Therefore, instead of a single QP, a series of QPs with increasing number of constraints should be solved. However, the number of constraints in each QP is usually very small and the problem can be solved almost instantly.

While a full theoretical analysis of the OSA attack is beyond the scope of this paper, we empirically observe that it substantially reduces computational cost. In practice, the number of effective constraints generated by the online procedure is far smaller than $T$—often resembling $\mathcal{O}(\log T)$—which leads to significantly faster attack construction compared to both full-trajectory and trajectory-free methods. Pseudocode for all methods is provided in Appendix K. Some interesting and fundamental questions arise regarding the attack designs discussed above:

- **Optimization formulation.** How can these attack designs be explicitly instantiated in real-world applications? In the next section, we show that for a linear reward model $r(\mathrm{X}) =$

$\mathrm{w}^\top \mathrm{X}$, all of the above formulations reduce to well-structured convex Quadratic Programs (QPs), which can be solved efficiently. In this setting, reducing the number of constraints substantially accelerates optimization. Interestingly, a similar argument also extends to wide neural network models.

- **Feasibility.** The efficiency gains above are only meaningful if the optimization problems are feasible, i.e., if there exists at least one perturbation vector $\boldsymbol{\delta}$ satisfying all constraints. In Section 3.2, we provide theoretical guarantees showing that, under mild assumptions on the reward model and the data-generating distributions $P_1, \ldots, P_K$, the constraint set is feasible with probability 1, provided that the data dimension $d$ (or, in the case of neural networks, the maximum width $\max_i W_i$) exceeds the number of constraints. Moreover, we establish a second main result: the norm of a feasible full-trajectory attack decreases at least as $\widetilde{O}(d^{-1/2})$ with dimension $d$.

## 3.1 OPTIMIZATION

The following result shows that, under a linear reward model, the proposed attack formulations admit an efficient convex optimization representation.

**Theorem 3.1** (Linear Reward Model). *Consider the three attack designs in Section 3 under the linear reward model $r(\mathrm{X}) = \mathrm{w}^\top \mathrm{X}$ for some fixed vector $\mathrm{w} \in \mathbb{R}^d$. Then there exist an indexed vector set $\{\mathrm{T}_{i,t}\}$ and a scalar set $\{R_{i,t}\}$, for $i \in [K-1]$ and $K < t \leq T$, such that all three attack designs can be expressed as the quadratic program*

$$\boldsymbol{\delta}^* \triangleq \underset{\boldsymbol{\delta}}{\arg\min} \ \|\boldsymbol{\delta}\|_2^2 \quad \text{subject to} \quad \boldsymbol{\delta}^\top \mathrm{T}_{i,t} > R_{i,t}, \quad \forall (i,t) \in \mathcal{I} \subset [K-1] \times [T]. \quad (4)$$

*The values of $\mathrm{T}_{i,t}$ and $R_{i,t}$ are determined solely by the offline datasets $\mathcal{D}_1, \ldots, \mathcal{D}_K$ and the original reward model $\mathrm{w}$. The joint arm–timestep index set $\mathcal{I}$ depends only on the choice of attack type (full-trajectory, trajectory-free, or OSA) and is determined according to the specification of the method.*

Proof is given in Appendix B. The computational cost is governed by the size of $\mathcal{I}$. In particular, for full-trajectory attacks we have $|\mathcal{I}| = \mathcal{O}(TK)$, for trajectory-free attacks $|\mathcal{I}| = \mathcal{O}(T)$, and for OSA attacks the effective size of $\mathcal{I}$ is typically much smaller in practice. Solving a QP of the form equation 4 with $d$ variables and $|\mathcal{I}|$ constraints requires approximately $\widetilde{\mathcal{O}}\big(|\mathcal{I}|^3 + d|\mathcal{I}|^2\big)$ operations (Gay et al., 1998), highlighting the central role of the constraint set size in the overall complexity. In our experiments (see Figure 7), we observed that OSA can reduce $|\mathcal{I}|$ to as small as $\mathcal{O}(\log T)$.

When the reward model is a neural network $\mathrm{NN}_{\boldsymbol{\theta}}(\cdot)$, the UCB score $\Lambda_i(t; \boldsymbol{\delta})$ in equation 1 is non-linear in $\boldsymbol{\delta}$, and the convex reduction in Theorem 3.1 no longer holds. However, results from neural tangent kernel (NTK) theory (Jacot et al., 2018) show that for sufficiently wide networks with randomly initialized weights, the network behaves approximately linearly in its parameters. Formally,

$$\mathrm{NN}_{\boldsymbol{\theta} + \boldsymbol{\delta}}(\mathrm{X}) = \mathrm{NN}_{\boldsymbol{\theta}}(\mathrm{X}) + \nabla_{\boldsymbol{\theta}} \mathrm{NN}_{\boldsymbol{\theta}}(\mathrm{X})^\top \boldsymbol{\delta} + \mathcal{O}\big(W_{\max}^{-1}\big) \quad \forall \mathrm{X} \in \mathbb{R}^d, \quad (5)$$

where $W_{\max}$ denotes the network width. Thus, for highly overparameterized networks the reward model can be well approximated by an affine function of $\boldsymbol{\delta}$, yielding a quadratic program analogous to equation 4, with affine constraints. Recall that $\nabla_{\boldsymbol{\theta}} \mathrm{NN}_{\boldsymbol{\theta}}(\boldsymbol{X})$ represents the embedding of the input $\boldsymbol{X}$ in the neural tangent space of the network $\mathrm{NN}_{\boldsymbol{\theta}}$, centered at the parameter vector $\boldsymbol{\theta}$ (Jacot et al., 2018). In fact, the output of an overparameterized neural network, for any fixed input $\boldsymbol{X}$, asymptotically mimics a linear reward function with respect to the perturbation vector $\boldsymbol{\delta}$—essentially reducing to the purely linear case.

**Corollary 3.2** (Overparameterized Neural Networks). *For asymptotically wide neural networks, linearization via the NTK approximation reduces the attack design to a convex quadratic program with $D$ variables as in Theorem 3.1, where $D$ is the number of parameters in $\boldsymbol{\theta}$. This program can be solved in $\widetilde{\mathcal{O}}(|\mathcal{I}|^3 + D|\mathcal{I}|^2)$ operations.*

## 3.2 THEORETICAL GUARANTEES: FEASIBILITY AND ATTACK NORM

In this section, we present two sets of theoretical guarantees. The first, stated in Theorem 3.3, establishes that under mild "non-degeneracy" conditions on the data-generating distributions $P_1, \ldots, P_K$, and in a sufficiently high-dimensional regime (i.e., $d \geq TK$), there always exists an attack (with

probability one over the sampling of datasets $\mathcal{D}_i$) that can steer the bandit toward *any* desired trajectory.

**Theorem 3.3** (Feasibility Guarantee). *Assume the data-generating distributions $P_1, \ldots, P_K$ are non-singular, i.e.,*

$$\lambda_{\min}(\mathsf{Cov}(P_i)) > 0,$$

*where $\mathsf{Cov}(\cdot)$ denotes the $d \times d$ covariance matrix and $\lambda_{\min}(\cdot)$ the minimum eigenvalue. If $d > (T - K)(K - 1)$, then with probability 1 all optimization problems corresponding to the full-trajectory attack, the trajectory-free attack, and the OSA attack are feasible.*

The proof (Appendix B) relies on linear-algebraic arguments and anti-concentration properties of non-degenerate distributions. A stronger result, requiring more technical machinery, is given in Theorem 3.4, where we show that in the high-dimensional regime, and under stronger assumptions on $P_1, \ldots, P_K$, the magnitude of the attack decreases as $\widetilde{\mathcal{O}}(d^{-1/2})$ with dimension. We conjecture that an even more general result could be established for near-product measures, which cover a broader class of distributions, though this lies beyond the scope of the present work.

For concreteness, let $\boldsymbol{\mu}_i \triangleq \mathbb{E}_{P_i}[\boldsymbol{X}]$ for $i \in [K]$, and assume $\|\boldsymbol{\mu}_i\|_2 = 1$ for all $i$. For the linear reward model, we further assume $\|\boldsymbol{w}\|_2 = 1$, with $\boldsymbol{\mu}_{i^*}^\top \boldsymbol{w} = 1$ and $\boldsymbol{\mu}_i^\top \boldsymbol{w} = 0$ for $i \neq i^*$. Such a configuration arises naturally in high-dimensional spaces, where independently drawn random vectors tend to be nearly orthogonal. We emphasize that these assumptions are introduced solely to simplify notation and exposition in subsequent sections; they are not essential to our analysis and do not affect the algorithmic design.

**Theorem 3.4** ($\ell_2$-norm of High-dimensional Attack). *Consider the assumptions mentioned above for $d > 2$. For simplicity, assume all $P_1, \ldots, P_K$ are product measures and $\mathsf{Cov}(P_i) = \boldsymbol{I}$ for all $i \in [K]$. Then, provided $d \geq KT$, the full-trajectory attack of equation 2 satisfies*

$$\|\boldsymbol{\delta}^*\|_2 \leq \mathcal{O}\left(\sqrt{\frac{T^3 \log T \cdot \log d}{Kd}}\right), \tag{6}$$

*with probability at least $1 - 2d^{-1}$ over the randomness of $\mathcal{D}_1, \ldots, \mathcal{D}_K$.*

The full proof, deferred to Appendix B, leverages tools from linear algebra, optimization, and random matrix theory. Here we provide a simplified proof sketch illustrating why attacks on high-dimensional bandits are feasible. It is well known that estimating the *mean* of high-dimensional random vectors becomes increasingly difficult as the dimension grows: fluctuations across many coordinates accumulate, making accurate estimation unreliable. A multi-armed bandit can be viewed as repeatedly estimating the means of several high-dimensional distributions and then selecting the arm that maximizes its reward. When the number of observations per arm, roughly $T/K$, is much smaller than the dimension $d$, such estimates are inherently unstable. In this regime, even a simple manipulation of the reward function—when it has sufficient degrees of freedom, as in linear models or overparameterized neural networks—can easily mislead the bandit. Theorem 3.4 further establishes that the required attack magnitude actually *decreases* as the dimension increases.

## 4 EXPERIMENTS

In this section, we conduct a series of experiments to assess the effectiveness of our proposed attack method. Our study focuses on some central questions: (**Q1**) How does the attack's performance differ between linear and non-linear reward models with synthetic data? (Section 4.1). (**Q2**) What is the impact of dimensionality on attack performance? (Section 4.2). (**Q3**) Can our method be applied to real-world settings, including real data and reward models? (Section 4.3). (**Q4**) Can our attack be applied to bandit algorithms other than UCB? (Section 4.4). (**Q5**) Are there any defenses that can prevent this attack under certain circumstances? (Section 4.5). (**Q6**) Can random noise perturbations achieve results comparable to our method? (Section 4.6). (**Q7**) How does increasing the width of the non-linear reward model affect the attack success rate? (Section 4.7). **Q8** How effective is our method against robust stochastic bandit algorithms under reward corruption? (Section 4.8)

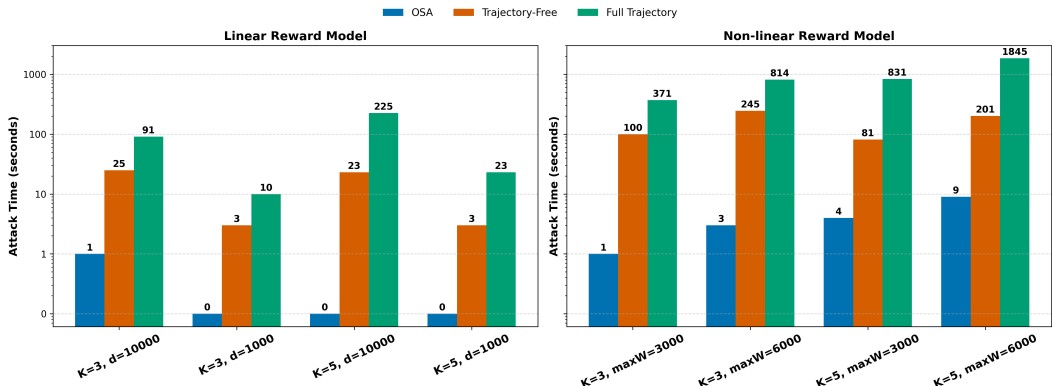

Figure 1: Comparison of attack performance on linear and nonlinear reward models using *OSA*, *Trajectory-Free*, and *Full Trajectory* methods. All attacks achieve a 100% success rate. The attack times (in seconds) are displayed on a logarithmic scale in the plot, highlighting the superior efficiency of *OSA* and illustrating the effect of varying parameters $K$ and $d$ (or $\max_i W_i$) on attack duration.

## 4.1 SYNTHETIC SETTING: ATTACK PERFORMANCE ACROSS REWARD MODELS

In this experiment, we evaluate the performance of our attack on reward models, examining its effectiveness in both linear and non-linear settings for synthetic data. To provide a more comprehensive assessment, we vary key parameters such as $K$ and $d$. Furthermore, We evaluate all variants of our method: *OSA* , *Trajectory-Free*, and *Full-Trajectory*.

This experiment focuses on two aspects: (1) demonstrating that both linear and non-linear reward models can be successfully attacked, and (2) comparing the attack time of the *OSA* and *Trajectory* methods. As shown in Figure 1, all variants of our attack achieve a 100% attack success rate ($ASR$) on the UCB algorithm. From these results, we observe that the *OSA* method substantially reduces the time required to generate perturbations (shown in the logarithmic scale). This demonstrates the clear efficiency advantage of the *OSA* approach. We also present additional experiments in Appendix C, showing that the number of constraints in *OSA* scales as $O(\log T)$.

## 4.2 IMPACT OF DIMENSIONALITY ON ATTACK PERFORMANCE

In this experiment, we validate our claim that increasing input dimensionality makes the bandit algorithm more vulnerable to attacks. To investigate this, we vary the input dimensionality while keeping all other parameters fixed. We observe that both the $\ell_2$ and $\ell_\infty$ norms of the perturbation decrease rapidly, indicating that in higher-dimensional settings, smaller perturbations suffice to successfully attack the algorithm. Additionally, we reformulate the optimization problem from a Feasibility Program (FP) to a Quadratic Program (QP) to evaluate how effectively our method approximates the optimal solution. In the low-dimensional regime, the *OSA* solution shows a small gap relative to the *Optimal* solution. However, as the input dimensionality increases, this gap diminishes, demonstrating that our *OSA* method achieves near-optimal attacks in higher-dimensional settings. We can conclude that our *OSA* method not only identifies a perturbation quickly, but also produces one that is very close to the optimal solution. Moreover, we investigate whether our attack in high-dimension is imperceptible or not based on its geometry in Appendix H.

## 4.3 REAL-WORLD SETTING: REAL DATA AND REWARD MODELS

We demonstrate that our attack can be applied in real-world settings, including real data and reward models, using image generative models as a case study. Specifically, we evaluate our attack on five generative models: Stable Diffusion 3 (SD3) and Stable Diffusion 1.4 (Rombach et al., 2022), Kandinsky 3 (Vladimir et al., 2024), Openjourney, and Stable Diffusion XL (Podell et al., 2023). We sample 30 random prompts from GenAI-Bench (Li et al., 2024) and curate logged data for each prompt by generating outputs from the models using seeds ranging from 1 to 100. For

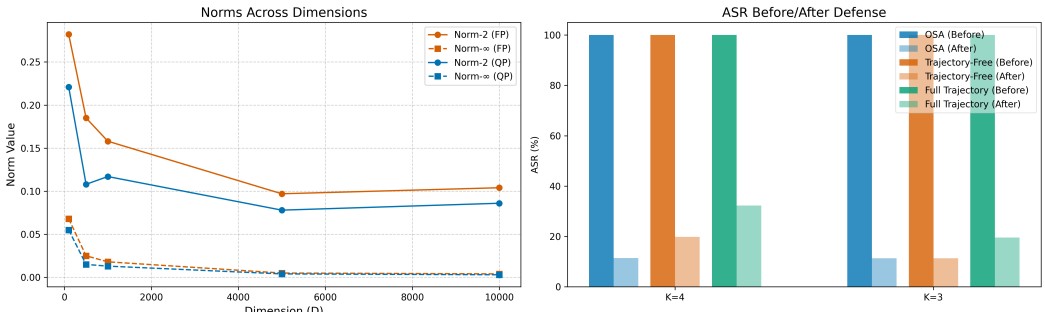

Figure 2: **Left:** Effect of increasing input dimensionality on attack magnitude. Both the $\ell_2$ and $\ell_\infty$ norms of the perturbation decrease as the input dimensionality increases, indicating that higher-dimensional inputs are more vulnerable to attacks. Experimental settings: $T = 100$, $K = 3$, ASR$= 100\%$. **Right:** Effect of applying the proposed defense on attack success rates. Shuffling a portion of the logged data before running the bandit algorithm significantly reduces the effectiveness of attacks. Specifically, we shuffle the first $T/2$ of the logged data before execution.

reward models, we use two widely adopted models: Image Reward (Xu et al., 2023), which evaluates the compositional alignment between the generated image and its prompt, and the Aesthetic Model (LAION-AI, 2021), which assesses the aesthetic quality of the generated image. Both models are widely used in prior work, and their weights are publicly available on HuggingFace. Moreover, both models are large, so we keep their weights frozen, except for a small subset used in the attack. As shown in Figure 3, the distribution of our attack success rate (ASR) is high: if we select a random prompt, generate logged data for it, and perform our attack, there is approximately an 80% probability that the ASR falls between 90–100%. We further demonstrate that our method can also attack a randomly initialized Reward model with arbitrary weights; details are provided in Appendix F. Additionally, to illustrate our method more clearly, a complete and detailed visualization of our attack on the Aesthetic Reward Model is provided in Appendix J.

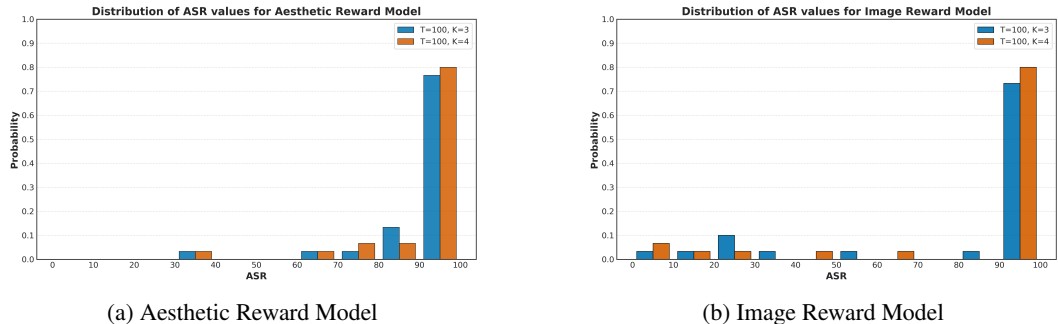

(a) Aesthetic Reward Model                    (b) Image Reward Model

Figure 3: Distribution of ASR values for the Aesthetic and Image reward models under the OSA method across two configurations. Each subplot shows the ASR distribution for one reward model.

## 4.4 ATTACKING BANDIT ALGORITHMS BEYOND UCB

Throughout this paper, we use the UCB (Auer, 2002) algorithm as our main bandit algorithm. To evaluate the robustness of our attack, we also consider scenarios where the adversary employs other bandit algorithms, specifically ETC (Garivier et al., 2016) and $\epsilon$-greedy (Dann et al., 2022). Table 1 presents the results of our attack on these algorithms, showing that it achieves a 100% ASR across all these algorithms. Additional details about these algorithms are provided in Appendix D.

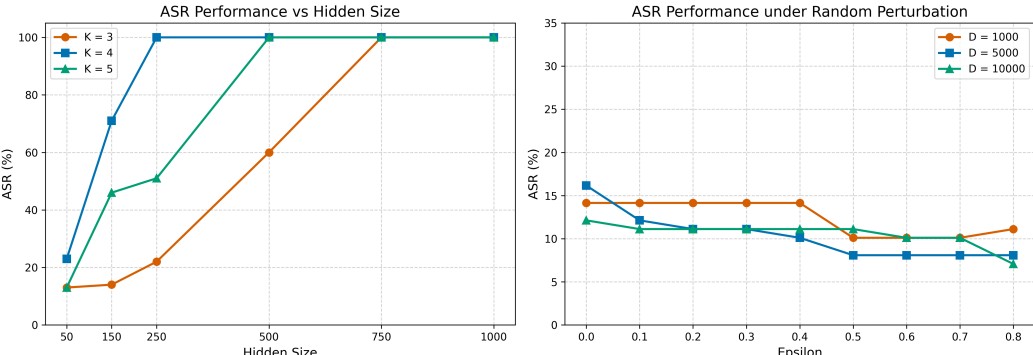

Figure 4: **Left**: Attack success rate as a function of the hidden layer width in the reward model ($T = 100$). The results show that a sufficiently wide hidden layer is required for the attack to succeed; once the width exceeds 750 neurons, the ASR reaches 100%. **Right**: Attack success rate versus the $\ell_2$ norm of random noise perturbations. The results show that even with increasing perturbation magnitude, the ASR remains nearly constant, demonstrating that random noise is largely ineffective in attacking the bandit algorithm.

### 4.5 OUR PROPOSED DEFENSE

We propose a defense that remains effective even when the environment is aware of the attack method, has access to the logged data, and is able to shuffle it. The defense operates by randomly shuffling a portion of the logged data before running the bandit algorithm. This simple yet effective strategy can substantially reduce the attack's success. As shown in Figure 2, shuffling only $T/2$ of the logged data is sufficient to significantly mitigate the attack. Additional results exploring other fractions of shuffled data are provided in Appendix G.

### 4.6 EFFECT OF RANDOM NOISE PERTURBATIONS ON ATTACK PERFORMANCE

We demonstrate that applying a random noise perturbation, even with a larger $\ell_2$ norm, fails to produce effective attacks. As shown in the right part of Figure 4, when no perturbation is applied, the attack success rate (ASR) is approximately 25%. Increasing the $\ell_2$ norm of the random perturbation has little impact on the ASR, which remains nearly constant across a wide range of perturbation magnitudes. This indicates that simple random noise is largely ineffective at compromising the bandit algorithm.

### 4.7 EFFECT OF NON-LINEAR REWARD MODEL WIDTH ON ATTACK SUCCESS

In this experiment, we investigate the impact of the reward model's width on attack performance. For our theoretical assumptions to hold in practice, the network's linear approximation must be valid. To achieve this, the reward model is designed with a single hidden layer of sufficient width. As the number of neurons increases, the linear approximation becomes more accurate (Jacot et al., 2018). Our results indicate that having enough neurons in this hidden layer is critical for a successful attack. As illustrated in the left part of Fig. 4, once the number of neurons exceeds 750, the attack success rate reaches 100%. Implementation details of the reward model training for synthetic data are provided in Appendix E.

### 4.8 COMPARISON OF OUR METHOD WITH ROBUST STOCHASTIC BANDIT ALGORITHMS

Existing corruption-robust MAB works (Niss & Tewari, 2020; Lykouris et al., 2018; Gupta et al., 2019) assume an adversary who actively modifies the realized rewards during the bandit interaction, i.e., the adversary is active, present throughout training, and directly alters the reward feedback as it is generated. In contrast, our attack perturbs the internal weights of the reward model before bandit training begins. The adversary does not have access to the environment where the bandit is run, which we believe is a much more realistic setting. However, we additionally evaluated

our attack against the Fast–Slow algorithm (Lykouris et al., 2018), a representative robust bandit method. As shown in Figure 5, this algorithm is also vulnerable. With a corruption parameter of $0.5$ (a typical setting in their own experiments), our attack succeeds $100\%$ of the time, while keeping the perturbation norm small (around $0.3$).

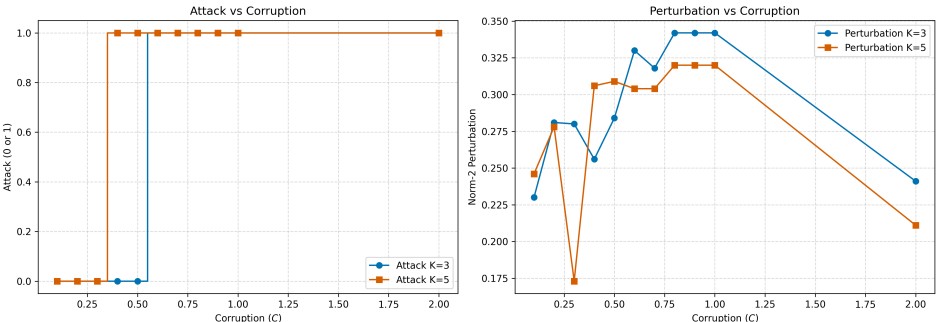

Figure 5: Our pre-training attack successfully hijacks the Fast–Slow robust bandit (Lykouris et al., 2018), achieving $100\%$ success with minimal $\ell_2$-norm perturbation ($\approx 0.3$) for $K \in \{3, 5\}$, $T = 1000$, and $d = 1000$.

Moreover, we evaluate our attack against the $\varepsilon$-contamination algorithm (Niss & Tewari, 2020), a robust variant of UCB. Figure 6 illustrates that for a reasonable contamination level ($\varepsilon = 0.15$), our attack achieves a $100\%$ ASR while requiring only a small $\ell_2$-norm perturbation (approximately $0.2$).

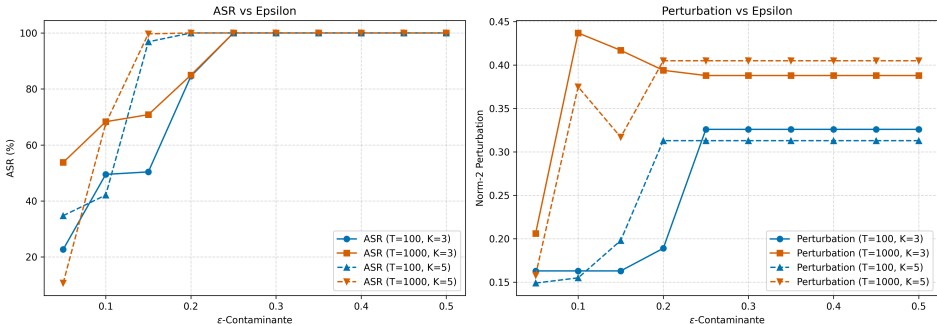

Figure 6: Effectiveness of our attack against the $\varepsilon$-contamination algorithm (Niss & Tewari, 2020), a robust UCB variant. Experiments are conducted with $T \in \{100, 1000\}$, $K \in \{3, 5\}$, $d = 1000$, $\alpha = 0.1$, $\sigma = 1$, using the $\alpha$-trimmed strategy.

## 5 CONCLUSION

In this work, we investigate the vulnerabilities of offline bandit algorithms used to evaluate machine learning models, including generative models. We show that publicly sharing reward model weights on platforms like Hugging Face exposes these models to adversarial attacks and highlight the critical role of logged data in offline bandit evaluation. We introduce a novel threat, demonstrating that even small, imperceptible perturbations to reward model weights can completely hijack bandit behavior. Our analysis starts with linear reward models, revealing their high susceptibility to manipulation in high-dimensional settings, and extends to nonlinear models such as ReLU-based neural networks. We validate our approach on real data by downloading publicly available weights from Hugging Face and successfully attacking these models. To mitigate such attacks, we propose a defense strategy and, for scenarios where the attacker only aims to prevent optimal arm selection, we introduce an efficient on–off heuristic that is both fast and effective. Finally, we provide theoretical support showing how perturbations can systematically steer bandit behavior in proportion to input dimensionality.

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

## A  RELATED WORK

While adversarial attacks on supervised learning models have been extensively studied (Szegedy et al., 2014; Moosavi-Dezfooli et al., 2017; Cicalese et al., 2020; Cohen et al., 2019; Dasgupta et al., 2019), the vulnerabilities of bandit algorithms remain less explored. This is particularly important in settings where reward models (Singh & Zheng, 2023; Cho et al., 2024; Hu et al., 2023; Hosseini et al., 2025) are combined with bandit algorithms as evaluators (Rezaei et al., 2025; Hu et al., 2024b), highlighting the need to understand prior adversarial attacks on bandit algorithms and to prepare for developing new attack strategies. Recent studies have investigated adversarial attacks on multi-armed bandits (MABs) (Jun et al., 2018; Liu & Shroff, 2019; Liu & Lai, 2020) and linear contextual bandits (Ma et al., 2018; Garcelon et al., 2020). In reward poisoning attacks, an adversary manipulates the reward signals received by the agent from the environment (Jun et al., 2018; Liu & Shroff, 2019; Niss & Tewari, 2020; Lykouris et al., 2018; Gupta et al., 2019; Hajiesmaili et al., 2020). By subtly altering these rewards, the attacker can bias the learning process, causing the agent to favor specific actions or arms, which may lead to suboptimal or adversary-driven behavior. In linear contextual bandits, adversarial attacks have focused on two main strategies. The first is reward poisoning, where the adversary modifies the rewards associated with chosen actions to steer the agent toward particular arms (Ma et al., 2018; Garcelon et al., 2020). The second is context poisoning, in which the attacker alters the observed context vectors while leaving the rewards unchanged, misleading the agent into making suboptimal or attacker-desired decisions (Garcelon et al., 2020). Together, these attacks demonstrate the vulnerabilities of linear contextual bandits to both reward and context-based manipulations. Our work differs from prior studies in that, rather than directly attacking the rewards, we target the reward models themselves. This approach is particularly relevant in the emerging usage of bandit algorithms as evaluation metrics. We show that in high-dimensional settings, reward models are vulnerable, making it possible for an attacker to easily manipulate them and thereby hijack the behavior of the bandit algorithm.

## B  PROOFS

*Proof of Theorem 3.1.* We first establish the result for the case of full-trajectory attacks. The other two variants differ only in that they employ a subset of the $(T-K)(K-1)$ constraints; hence, both the optimization formulation and the feasibility properties can be deduced directly from the analysis of the full-trajectory case. In this regard, we define the residual outputs Z as $Z_j^{(i)} \triangleq X_j^{(i)} - \boldsymbol{\mu}_i$, therefore we have $\mathbb{E}[Z_j^{(i)}] = \mathbf{0}$ for all valid $i$ and $j$. Let us define $S_j^{(i)} \in \mathbb{R}^d$ for $i \in [K]$ and $k \in [n_i]$ as

$$S_k^{(i)} \triangleq \frac{1}{k} \sum_{j=1}^{k} X_j^{(i)} = \boldsymbol{\mu}_i + \frac{1}{k} \sum_{j=1}^{k} Z_j^{(i)} \,.$$

In this regard, we have $\mathbb{E}[S_k^{(i)}] = \boldsymbol{\mu}_i$, for all $i$ and regardless of $k \geq 1$. Therefore, the second sub-problem above reduces to

$$\arg\min_{\boldsymbol{\delta}} \| \boldsymbol{\delta} \|_2^2 \tag{7}$$
$$\text{subject to} \quad (\text{w} + \boldsymbol{\delta})^\top \left[ S_{N_i(t)}^{(i)} - S_{N_j(t)}^{(j)} \right] > \sqrt{2 \log t} \left( N_j^{-1/2}(t) - N_i^{-1/2}(t) \right)$$
$$(\text{for } i = \widetilde{A}_t, \ \forall j \neq i, \ K < t \leq T),$$

and $N_i(t) \triangleq \sum_{t'=1}^{t-1} \mathbb{1}(\widetilde{A}_{t'} = i)$. The above set of inequality constraints can be further simplified via introducing new symbols, and hence the above convex feasibility problem can be rewritten as

$$\arg\min_{\boldsymbol{\delta}} \| \boldsymbol{\delta} \|_2^2 \tag{8}$$
$$\text{subject to} \quad \boldsymbol{\delta}^\top T_{j,t} > R_{j,t}, \quad \forall j \in [K-1], \ K < t \leq T,$$

where the vectors $\mathrm{T}_{j,t}$ and scalars $R_{j,t}$ depend on samples in datasets $\mathcal{D}_i$ and the target sequence $\widetilde{A}_t$. More precisely, we have

$$\mathrm{T}_{j,t} \triangleq (\boldsymbol{\mu}_i - \boldsymbol{\mu}_{j'}) + \left[ \frac{1}{N_i(t)} \sum_{l=1}^{N_i(t)} \mathrm{Z}_l^{(i)} - \frac{1}{N_{j'}(t)} \sum_{l=1}^{N_{j'}(t)} \mathrm{Z}_l^{(j')} \right], \tag{9}$$

$$R_{j,t} \triangleq -\mathrm{w}^\top \mathrm{T}_{j,t} + \sqrt{2 \log t} \left( N_{j'}^{-1/2}(t) - N_i^{-1/2}(t) \right),$$

where we have $i \triangleq \widetilde{A}_t$, and $j' \in [K-1]$ is the index of the set $[K]$ after removing $\widetilde{A}_t$.

In the case where the reward model is a neural network rather than linear, the NTK approximation in equation 5, together with the assumption $\max_i W_i \gg 1$, yields an analogous QP form, with modified definitions of $\mathrm{T}_{j,t}$ and $R_{j,t}$:

$$\mathrm{T}_{j,t} \triangleq \frac{1}{N_i(t)} \sum_{l=1}^{N_i(t)} \nabla_{\boldsymbol{\theta}} \mathrm{NN}_{\boldsymbol{\theta}}(\mathrm{X}_l^{(i)}) - \frac{1}{N_{j'}(t)} \sum_{l=1}^{N_{j'}(t)} \nabla_{\boldsymbol{\theta}} \mathrm{NN}_{\boldsymbol{\theta}}(\mathrm{X}_l^{(j')}), \tag{10}$$

$$R_{j,t} \triangleq - \left[ \frac{1}{N_i(t)} \sum_{l=1}^{N_i(t)} \mathrm{NN}_{\boldsymbol{\theta}}(\mathrm{X}_l^{(i)}) - \frac{1}{N_{j'}(t)} \sum_{l=1}^{N_{j'}(t)} \mathrm{NN}_{\boldsymbol{\theta}}(\mathrm{X}_l^{(j')}) \right] + \sqrt{2 \log t} \left( N_{j'}^{-1/2}(t) - N_i^{-1/2}(t) \right).$$

This completes the proof. $\qquad\square$

*Proof of Theorem 3.3.* Without loss of generality, we assume $i^* = 1$. Recalling the notation from the proof of Theorem 3.1, for all valid $j$ and $t$, we have

$$\mathbb{E}\left[\mathrm{w}^\top \mathrm{T}_{j,t}\right] = \begin{cases} 1 & \widetilde{A}_t = 1 \\ 0 & \widetilde{A}_t \neq 1, j \neq 1 \\ -1 & \widetilde{A}_t \neq 1, j = 1 \end{cases}, \tag{11}$$

$$\mathsf{Var}\left[\mathrm{w}^\top \mathrm{T}_{j,t}\right] = \frac{1}{N_i(t)} \mathrm{w}^\top \mathsf{Cov}_{P_i} \mathrm{w} + \frac{1}{N_{j'}(t)} \mathrm{w}^\top \mathsf{Cov}_{P_{j'}} \mathrm{w}.$$

Note that both $\mathrm{T}_{j,t}$ and $R_{j,t}$ are deterministic. Also, we have

$$|R_{j,t}| \leq \left| \mathrm{w}^\top \mathrm{T}_{j,t} \right| + \sqrt{2 \log t} \left( 1 - t^{-1/2} \right). \tag{12}$$

The set of constraints can be rewritten in the following matrix form:

$$\mathbb{T} \boldsymbol{\delta} \succeq \boldsymbol{R}, \tag{13}$$

where $C \times d$ matrix $\mathbb{T}$ and $C$-dimensional vector $\boldsymbol{R}$ are formed via row-wise concatenation of $\mathrm{T}_{j,t}$s and $R_{j,t}$s, respectively. Here, $C$ denotes the number of constraints which is at most $(K-1)(T-K)$. Note that we have replaced $\succ$ with $\succeq$ by assuming infinitesimally small margins in the constraints. This modification simplifies the subsequent part of the proof. Also, it should be noted that both $\mathbb{T}$ and $\boldsymbol{R}$ are random.

We need to show that the stochastic inequality system $\mathbb{T} \boldsymbol{\delta} \succeq \boldsymbol{R}$ is feasible with probability 1. We proceed by contradiction. Without loss of generality, assume that the first inequality $\mathrm{T}_1^\top \boldsymbol{\delta} \geq R_1$ is infeasible (with positive probability), given that the remaining $C-1$ inequalities are satisfied. Formally, assume

$$\mathbb{P}\left( \left\{ \boldsymbol{\delta} \mid \mathrm{T}_1^\top \boldsymbol{\delta} \geq R_1 \right\} \cap \left\{ \boldsymbol{\delta} \mid \mathrm{T}_i^\top \boldsymbol{\delta} \geq R_i, \ i = 2, \ldots, C \right\} = \varnothing \right) > 0.$$

Then we must have

$$\mathbb{P}\left( \left\{ \boldsymbol{\delta} \mid \mathbb{T} \boldsymbol{\delta} = \boldsymbol{R} \right\} = \varnothing \right) > 0,$$

that is, the stochastic equation system $\mathbb{T} \boldsymbol{\delta} = \boldsymbol{R}$ is unsolvable with positive probability. Since the number of variables $d$ exceeds the maximum number of equations $(T-K)(K-1)$, this can only occur if the rows of $\mathbb{T}$ fail to be linearly independent with positive probability.

However, this is impossible by construction of $\mathbb{T}$ in equation 9. Indeed, since all residuals $\{\mathrm{Z}_j^{(i)}\}_{j \in [n_i]}$ are independently drawn and each distribution $P_i$ is non-degenerate (i.e., $\lambda_{\min}(\mathsf{Cov}(P_i)) > 0$ for every $i \in [K]$), the rows of $\mathbb{T}$ are distributed in general position and are therefore linearly independent with probability 1. This completes the proof. $\qquad\square$

*Proof of Theorem 3.4.* We begin with the following lemma:

**Lemma B.1.** *Assume* $(T - K)(K - 1) < d$ *and consider the constrained optimization problem*

$$\boldsymbol{\delta}^* \triangleq \arg\min_{\boldsymbol{\delta}} \| \boldsymbol{\delta} \|_2^2 \quad \text{subject to} \quad \mathbb{T}\boldsymbol{\delta} \succeq \boldsymbol{R}. \tag{14}$$

*Define* $\widetilde{\boldsymbol{\delta}} \triangleq \mathbb{T}^\dagger \boldsymbol{R}$, *where* $\dagger$ *denotes the Moore–Penrose pseudo-inverse, i.e.,* $\widetilde{\boldsymbol{\delta}} = (\mathbb{T}\mathbb{T}^\top)^{-1}\mathbb{T}^\top \boldsymbol{R}$. *Then we have* $\| \boldsymbol{\delta}^* \|_2 \leq \| \widetilde{\boldsymbol{\delta}} \|_2$.

*Proof.* The argument is straightforward. Removing any constraint from equation 14 (equivalently, removing a row of $\mathbb{T}$) cannot increase $\| \boldsymbol{\delta}^* \|_2$. Hence, the maximum possible $\ell_2$-norm of $\boldsymbol{\delta}^*$ occurs when all constraints are active. Formally,

$$\| \boldsymbol{\delta}^* \|_2 \leq \left\| \arg\min_{\boldsymbol{\delta}} \| \boldsymbol{\delta} \|_2^2 \quad \text{subject to} \quad \mathbb{T}\boldsymbol{\delta} = \boldsymbol{R} \right\|_2$$

$$= \| \widetilde{\boldsymbol{\delta}} \|_2, \tag{15}$$

where the last equality follows from the definition of the Moore–Penrose pseudo-inverse, which yields the unique minimum-$\ell_2$-norm solution to an underdetermined linear system. Also, according to Theorem 3.3, we know the linear equation system $\mathbb{T}\boldsymbol{\delta} = \boldsymbol{R}$ is solvable (hence $\widetilde{\boldsymbol{\delta}}$ exists) with probability 1. This completes the proof. □

Next, we bound $\| \widetilde{\boldsymbol{\delta}} \|_2$. In order to do this, first note that

$$\| \widetilde{\boldsymbol{\delta}} \|_2 \leq \frac{\| \boldsymbol{R} \|_2}{\sigma_{\min}(\mathbb{T})}, \tag{16}$$

where $\sigma_{\min}(\cdot)$ denotes the minimum singular value. Therefore, we need to: i) find a high-probability bound for $\sigma_{\min}(\mathbb{T})$, and ii) establish a similar bound for $\| \boldsymbol{R} \|_2$. This requires an appropriate matrix representation for the construction of $\mathbb{T}$, whose rows are defined in equation 9:

$$\text{any row of } \mathbb{T} = \boldsymbol{\mu}_i - \boldsymbol{\mu}_j + \left[ \frac{1}{N_i(t)} \sum_{l=1}^{N_i(t)} \boldsymbol{Z}_l^{(i)} - \frac{1}{N_j(t)} \sum_{l=1}^{N_j(t)} \boldsymbol{Z}_l^{(j)} \right], \tag{17}$$

for any two properly defined indices $i, j$. We then leverage existing tools from random matrix theory to establish our main results.

**Matrix-form generation of $\mathbb{T}$.** Let us define the residual matrix $\Gamma$ as follows:

$$\Gamma \triangleq \left[ \boldsymbol{Z}_1^{(1)} \mid \cdots \mid \boldsymbol{Z}_{n_1}^{(1)} \mid \cdots \mid \boldsymbol{Z}_1^{(K)} \mid \cdots \mid \boldsymbol{Z}_{n_K}^{(K)} \right]^\top, \tag{18}$$

which is an $(n_1 + \cdots + n_K) \times d$ dimensional matrix that collects all the $\boldsymbol{Z}$ vectors as its rows. According to the assumptions in the statement of the theorem, $\Gamma$ is a random matrix with independent rows. Moreover, since each data-generating distribution is assumed to be a product measure, the entries in each row are also independent and have variance 1. We are particularly interested in a submatrix of $\Gamma$, denoted by $\Gamma^* \in \mathbb{R}^{(T-K) \times d}$, which consists only of those vectors $\boldsymbol{Z}_j^{(i)}$ that correspond to arms pulled during the target trajectory for time-steps $K < t \leq T$. Since the target trajectory is fixed and deterministic, $\Gamma^*$ is still a random matrix with independently distributed entries of unit variance.

Recall that $N_i(T)$ denotes the number of times arm $i$ has been pulled up to the horizon $T$ for a given fixed target trajectory. For any non-degenerate trajectory $\widetilde{A}_t$, we have $N_i(T) \leq \mathcal{O}(T/K)$, $\forall i \in [K]$. Let $m$ denote the maximum number of times any arm has appeared during the trajectory, so that $m \leq \mathcal{O}(T/K)$. Now, consider the following family of matrices:

$$\Lambda_n \triangleq \begin{bmatrix} 1 & 1/2 & \cdots & 1/n \\ 0 & 1/2 & \cdots & 1/n \\ \vdots & \ddots & \ddots & \vdots \\ 0 & 0 & \cdots & 1/n \end{bmatrix}, \quad \forall n \in \mathbb{N}. \tag{19}$$

**Lemma B.2.** *we have $\lambda_{\min}(\Lambda_n) = 1/n$.*

*Proof.* Proof is straightforward. For some fixed $n \in \mathbb{N}$, let us denote the eigenvalues of $\Lambda_n$ by $\lambda_1, \ldots, \lambda_n$. Then, we have $\det(\Lambda_n - \lambda_i \boldsymbol{I}) = 0$ for all $i \in [n]$. On the other hand, since $\Lambda_n - \lambda_i \boldsymbol{I}$ is upper-triangular, we have

$$\det(\Lambda_n - \lambda_j \boldsymbol{I}) = \prod_{i=1}^{n} \left( \frac{1}{i} - \lambda_j \right), \quad \forall j \in [n].$$

Therefore, we must have $\lambda_j = 1/j$ for all $j \in [n]$ and consequently $\lambda_{\min}(\Lambda_n) = 1/n$, which completes the proof. $\square$

Consider the matrix $\Lambda_m$, and let us denote its columns by $\boldsymbol{v}_1, \ldots, \boldsymbol{v}_m \in \mathbb{R}^m$. Based on the definition of $\mathbb{T}$ in equation 9, each row of $\mathbb{T}$ consists of a mean-difference term $\boldsymbol{\mu}_i - \boldsymbol{\mu}_j$ (for properly defined indices $i, j$), together with a term that depends solely on residuals. This residual part can be rewritten as

$$\frac{1}{N_i(t)} \sum_{l=1}^{N_i(t)} \boldsymbol{Z}_l^{(i)} - \frac{1}{N_j(t)} \sum_{l=1}^{N_j(t)} \boldsymbol{Z}_l^{(j)} = \left( \begin{bmatrix} \boldsymbol{0} \mid \boldsymbol{v}_{N_i(t)} \mid \boldsymbol{0} \end{bmatrix} - \begin{bmatrix} \boldsymbol{0} \mid \boldsymbol{v}_{N_j(t)} \mid \boldsymbol{0} \end{bmatrix} \right)^{\top} \Gamma^*, \quad (20)$$

where $\begin{bmatrix} \boldsymbol{0} \mid \boldsymbol{v}_i \mid \boldsymbol{0} \end{bmatrix}$ denotes a vector of dimension $T - K$ obtained by padding $\boldsymbol{v}_i$ with zeros both before and after. In this regard, the random matrix $\mathbb{T}$ can be expressed as

$$\mathbb{T} = \boldsymbol{M} + \boldsymbol{V}^{\top} \Gamma^*, \quad (21)$$

where $\boldsymbol{M}$ is a low-rank matrix consisting of the mean-difference terms $\boldsymbol{\mu}_i - \boldsymbol{\mu}_j$ as its rows. This includes (potentially) all possible pairwise differences between the means of the $K$ distributions $P_1, \ldots, P_K$, with repetitions as required. Note that $\boldsymbol{M}$ can have at most $K$ nonzero singular values. The matrix $\boldsymbol{V} \in \mathbb{R}^{(T-K) \times (T-K)(K-1)}$ collects all combinations of vectors of the form

$$\begin{bmatrix} \boldsymbol{0} \mid \boldsymbol{v}_{N_i(t)} \mid \boldsymbol{0} \end{bmatrix} - \begin{bmatrix} \boldsymbol{0} \mid \boldsymbol{v}_{N_j(t)} \mid \boldsymbol{0} \end{bmatrix},$$

as its columns. The matrix $\boldsymbol{V}$ is fully determined by the target trajectory and can be regarded as a row/column-wise augmentation of $\Lambda_m$. Therefore,

$$\sigma_{\min}(\boldsymbol{V}) \geq \mathcal{O}(\sigma_{\min}(\Lambda_m)) = \mathcal{O}(1/m) \geq \mathcal{O}(K/T). \quad (22)$$

Consequently, we obtain

$$\sigma_{\min}(\mathbb{T}) \geq \mathcal{O}(K/T) \cdot \sigma_{\min}(\Gamma^*). \quad (23)$$

What remains, with respect to bounding the denominator in equation 16, is to establish a high-probability bound on $\sigma_{\min}(\Gamma^*)$. To this aim, we invoke a well-established result from random matrix theory. There are several known high-probability bounds for the minimum and maximum singular values of random matrices with independently drawn entries of equal variance 1. In particular, we have:

**Theorem B.3** (Based on Theorem 6.1 of Wainwright (2019)). *For any positive $\delta$, and assuming $d \gg T - K$, we have*

$$\sigma_{\min}(\Gamma^*) \geq \sqrt{d}(1 - \delta) - \sqrt{T - K}, \quad (24)$$

*with probability at least $1 - e^{-d\delta^2/2}$.*

The proof can be found in the reference. Therefore, for any $\delta > 0$,

$$\mathbb{P}\left( \sigma_{\min}(\mathbb{T}) \geq \mathcal{O}\left( \frac{K\sqrt{d}}{T} \right) \left( 1 - \delta - \sqrt{\tfrac{T-K}{d}} \right) \right) \geq 1 - e^{-d\delta^2/2}. \quad (25)$$

Here, we used the fact that data-generating distributions are product measures with independent entries, such as $\mathcal{N}(\boldsymbol{\mu}, \sigma^2 \boldsymbol{I})$ for any mean vector $\boldsymbol{\mu}$ and variance-per-dimension $\sigma^2$. Next, we try to (with high probability) upper-bound the nominator of equation 16.

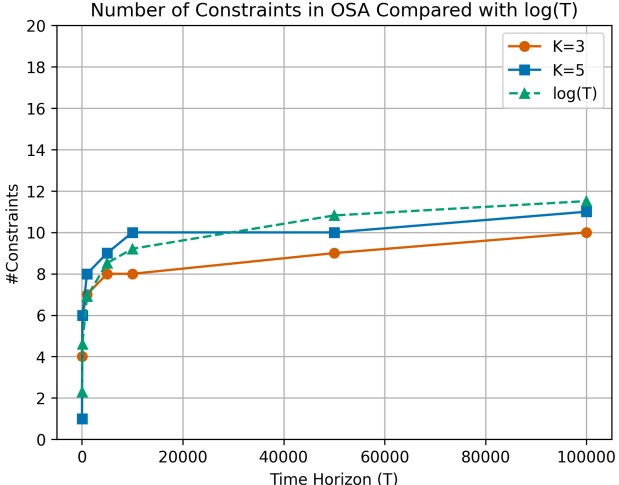

Figure 7: Number of constraints in the *OSA* method as a function of the time horizon $T$. The results show that the number of constraints grows logarithmically with $T$ ($O(\log T)$) for both $K = 3$ and $K = 5$ arms, explaining the high efficiency of the method.

**High-probability bounds on $\|\boldsymbol{R}\|_2$.**   What remains is to bound $\|\boldsymbol{R}\|_2$ in equation 16. From equation 11 and equation 12, and using Chernoff bound (Wainwright, 2019), we obtain a naive high-probability bound as follows:

$$\mathbb{P}\Big(|R_i| \geq c^{1/2}\,\mathcal{O}\big(1 + \sqrt{\log T}\big)\Big) \leq e^{-c}, \quad \forall\, c > 0,\ i \in [(K-1)(T-K)]. \tag{26}$$

Therefore, using $(T - K)(K - 1) \leq KT$ and assuming $T \gg 1$, we have

$$\mathbb{P}\Big(\|\boldsymbol{R}\|_2 \geq \mathcal{O}\big(\sqrt{cKT \log T}\big)\Big) \leq e^{-c}, \quad \forall\, c > 0. \tag{27}$$

Combining equation 25 and equation 27, and applying the union bound, we obtain the following high-probability bound for $\|\widetilde{\boldsymbol{\delta}}\|_2$ (and hence $\|\boldsymbol{\delta}^*\|_2$), for any $\delta, c > 0$:

$$\mathbb{P}\left(\|\widetilde{\boldsymbol{\delta}}\|_2 \leq \mathcal{O}\Big(\frac{T^{3/2}\sqrt{\log T}}{\sqrt{Kd}}\Big) \cdot \frac{\sqrt{c}}{1 - \delta - \sqrt{(T-K)/d}}\right) \geq 1 - e^{-c} - e^{-d\delta^2/2}. \tag{28}$$

Finally, choosing $c = \log d$ and $\delta = d^{-1/3}$ yields the bound stated in the theorem and completes the proof. $\qquad\square$

## C   ANALYSIS OF CONSTRAINT NUMBER IN THE OSA METHOD

We study the effect of increasing the time horizon $T$ on the number of constraints in the *OSA* method. Our observations show that, while the number of constraints grows with $T$, this growth follows an $O(\log T)$ pattern, as illustrated in Fig. 7 for $T$ ranging from 10 to 100,000. To confirm this behavior, we repeat the experiment for two values of the number of arms, $K = 3$ and $K = 5$, and observe the same pattern in both cases. This small number of constraints is one of the main reasons that makes the *OSA* method so fast.

## D   ADDITIONAL DETAILS ON ETC AND $\epsilon$-GREEDY ALGORITHMS

### D.1   ETC ALGORITHM

The Explore-Then-Commit (ETC) (Garivier et al., 2016) algorithm is a simple bandit strategy that divides the learning process into two distinct phases. During the exploration phase, the algorithm

Table 1: Comparison of attack performance on linear and non-linear reward models using the $\epsilon$-*greedy* and *ETC* methods. All attacks achieve a $100\%$ success rate ($ASR = 100$). The table illustrates how varying the parameters $K$ and $D$ (or $\max_i W_i$) affects the attack duration. In this experiment, we set $T = 100$ and the number of exploration rounds for *ETC* to $m = 5$.

| Reward Models | Linear Reward Model | | | Non-linear Reward Model | | |
|---|---|---|---|---|---|---|
| | $K$ | $d$ | Attack Time | $K$ | $\max_i W_i$ | Attack Time |
| **$\epsilon$-greedy Attack** | | | | | | |
| | 3 | 10000 | 00:00 | 3 | 3000 | 00:22 |
| | 3 | 1000 | 00:00 | 3 | 6000 | 00:12 |
| | 5 | 10000 | 00:00 | 5 | 3000 | 00:18 |
| | 5 | 1000 | 00:00 | 5 | 6000 | 00:13 |
| **Explore-Then-Commit (ETC)** | | | | | | |
| | 3 | 10000 | 00:00 | 3 | 3000 | 00:03 |
| | 3 | 1000 | 00:00 | 3 | 6000 | 00:06 |
| | 5 | 10000 | 00:00 | 5 | 3000 | 00:04 |
| | 5 | 1000 | 00:00 | 5 | 6000 | 00:09 |

uniformly samples each arm a predetermined number of times to gather initial reward estimates. Once the exploration phase is complete, ETC commits to the arm with the highest estimated reward for the remainder of the time horizon. This approach provides a balance between exploration and exploitation while remaining easy to implement and analyze. The length of the exploration phase can be tuned to optimize performance depending on the problem setting.

To attack ETC, the adversary only needs to wait until the exploration phase is complete. At the transition between the exploration and commit phases, the adversary manipulates the rewards so that the empirical mean of the target arm exceeds that of the optimal arm. During the subsequent commit phase, the algorithm then selects the target arm. Since the ETC constraint set involves only a single constraint, the attack is fast and straightforward, particularly in high-dimensional settings.

### D.2 $\epsilon$-GREEDY ALGORITHM

The $\epsilon$-greedy (Dann et al., 2022) algorithm selects a random arm with probability $\epsilon$ (exploration) and the arm with the highest estimated reward with probability $1 - \epsilon$ (exploitation) at each time step. In our experiments, we use an adaptive $\epsilon$ that decreases over time, allowing the algorithm to explore more during the early stages and gradually exploit the best-known arm as more information is gathered. This time-dependent strategy enables the algorithm to balance exploration and exploitation effectively, and the choice of the initial $\epsilon$ and its decay schedule directly influences its performance.

To attack $\epsilon$-greedy, the adversary follows the same principle as *OSA*. During random arm selections, no attack is performed. However, when the algorithm selects the arm with the highest empirical mean, the adversary intervenes to prevent the optimal arm from being pulled. Specifically, the target arm is chosen as the runner-up, and its empirical value is increased to surpass the optimal arm, as modifying the runner-up is simpler than other arms. This approach makes the attack fast and highly efficient.

For the ETC algorithm, we set the number of exploration rounds to $m = 5$, meaning each arm is selected exactly five times before the commit phase begins. For the $\epsilon$-greedy algorithm, we set the initial $\epsilon$ to $0.1$ and apply decay over time, with a minimum value of $0.01$.

## E IMPLEMENTATION DETAILS OF REWARD MODEL TRAINING FOR SYNTHETIC DATA

We train several neural networks, each with a single hidden layer whose size varies depending on the task. The networks are optimized using Adam (Adam et al., 2014) and employ the ReLU activation function (Agarap, 2018). Training is performed using the mean squared error (MSE) loss. The

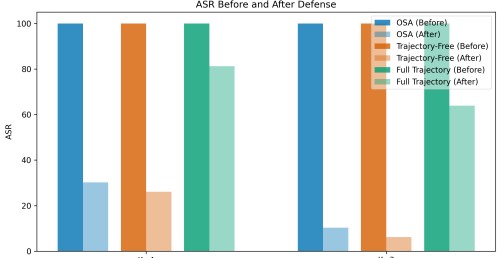 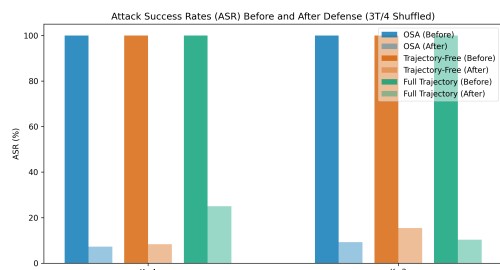

(a) ASR before and after defense on the linear reward model when $T/4$ of the logged data is shuffled. The defense significantly reduces the attack success rates across all attack methods.

(b) ASR before and after defense on the linear reward model when $3T/4$ of the logged data is shuffled. The defense further reduces attack success rates compared to the $T/4$ case.

Figure 8: Comparison of ASR before and after defense on the linear reward model under different fractions of logged data shuffled as a defense.

logged data is split into training and validation sets, with 80% used for training and 20% reserved for validation. Networks are trained for 100 epochs with a batch size of 1024 and a learning rate of $10^{-3}$. For supervision, the ground truth is computed as the inner product of the data with the mean of the optimal arm.

## F    ADDITIONAL RESULTS ON ATTACKING THE REAL DATASET

To demonstrate the generality of our attack, in addition to two pretrained reward models commonly used to evaluate generative image models, Image Reward (Xu et al., 2023) and Aesthetic Model (LAION-AI, 2021), we attack a randomized reward network. This random reward model is initialized with random weights and we freeze the majority of its parameters except for a single hidden layer. Despite its random initialization, the model's trajectory can still be hijacked by our method: the ASR distribution for this random reward model closely matches those of the Image Reward and Aesthetic models in Figure 3. Figure 9 shows the distribution of our attack on the random reward model.

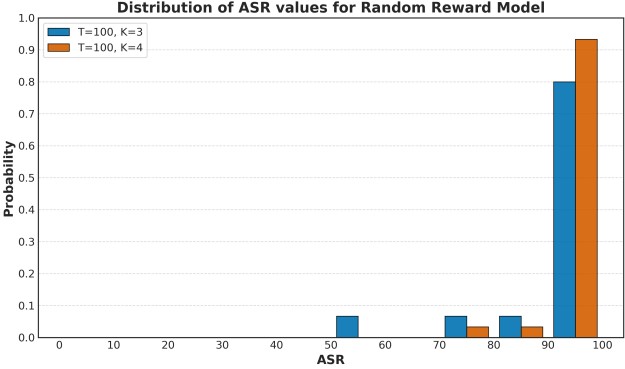

Figure 9: ASR distribution on the random reward model, showing vulnerability similar to the Image Reward and Aesthetic models despite random weights.

## G    ADDITIONAL RESULTS ON THE DEFENSE METHOD

The effectiveness of the shuffling defense can be further evaluated by varying the fraction of logged data that is shuffled. In particular, we consider two additional settings where $T/2$ and $3T/4$ of the logged data are shuffled before running the bandit algorithm. The results demonstrate that increasing

the portion of shuffled data substantially reduces the attack success rate (ASR) across all attack methods. Detailed results for these settings are presented in Figure 8.

# H    GEOMETRY OF ATTACK VECTOR

We visualize the geometry of the attack vector relative to other vectors, we reduce the perturbation space to three dimensions using PCA. As shown in Fig. 10a and Fig. 10b, the perturbations are neither trivial nor simply the reverse of the vector $w$. This demonstrates that the attack is subtle, imperceptible, and not easily detectable from the bandit algorithm's perspective.

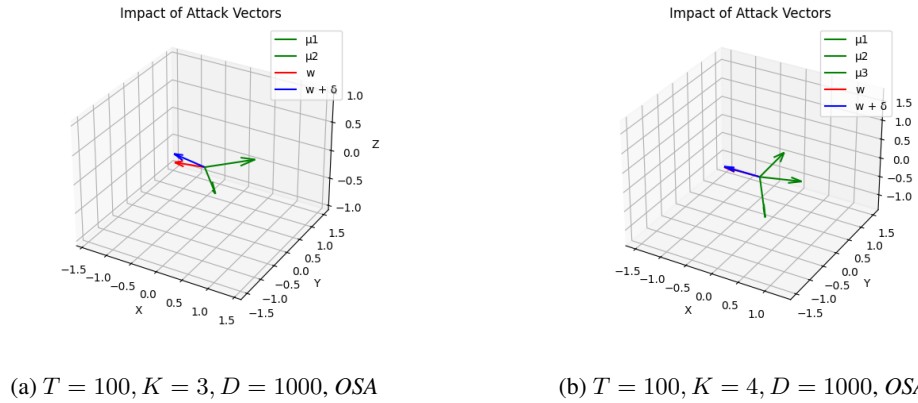

(a) $T = 100, K = 3, D = 1000, OSA$          (b) $T = 100, K = 4, D = 1000, OSA$

Figure 10: 3D visualization of perturbations for different values of $K$.

# I    A REALISTIC AND PRACTICAL THREAT MODEL FOR MODERN BANDIT SYSTEMS

Our setting, similar to most modern and practical MAB formulations, assumes that pulling an arm produces a data sample, not a direct reward value, and the bandit/learner then evaluates the reward based on that sample. For example, an image generator or an LLM returns an image or text when queried (equivalent to pulling an arm), and the learner assesses a reward (e.g., fidelity or quality) using some reward function. Such reward functions may be linear or neural networks. Here, sample generation is handled by the environment, to which we assume no access, whereas reward functions are typically derived from publicly available pretrained models that can be poisoned and later downloaded by an unknowing learner for bandit training. Several of the cited works (Rezaei et al., 2025; Hu et al., 2024b) in our Introduction highlight exactly this kind of setting. Another point to mention is that we believe our setting is more realistic in practice than previous attack strategies. Instantiated attack perturbation requires the adversary to be online, active, and present inside the private server where the bandit is trained. This is actually too much to assume from the adversary's side. We (both theoretically and experimentally) show that poisoning the reward function *before* training even begins can still cause equally significant damage, while the attack itself is much simpler to perform. We believe it is important for the community to recognize that such a vulnerability exists in offline bandits, which may motivate future efforts toward developing defense mechanisms.

# J    VISUALIZATION OF OSA METHOD ON REAL REWARD MODEL

Figure 11 illustrates our OSA attack on the Aesthetic Reward Model (LAION-AI, 2021). The first panel presents the baseline experiment: UCB (Auer, 2002) is run using the reward scores from the model, which leads it to consistently select SD3 (Rombach et al., 2022) as the optimal arm since it achieves the highest aesthetic scores. The plot depicts the arm-selection trajectory over time under this setting. The second panel shows the same experiment under our attack. By manipulating the weights of reward model, the attacker hijacks UCB so that it selects the Openjourney model as the optimal arm. Timesteps targeted by OSA method are highlighted in gray.

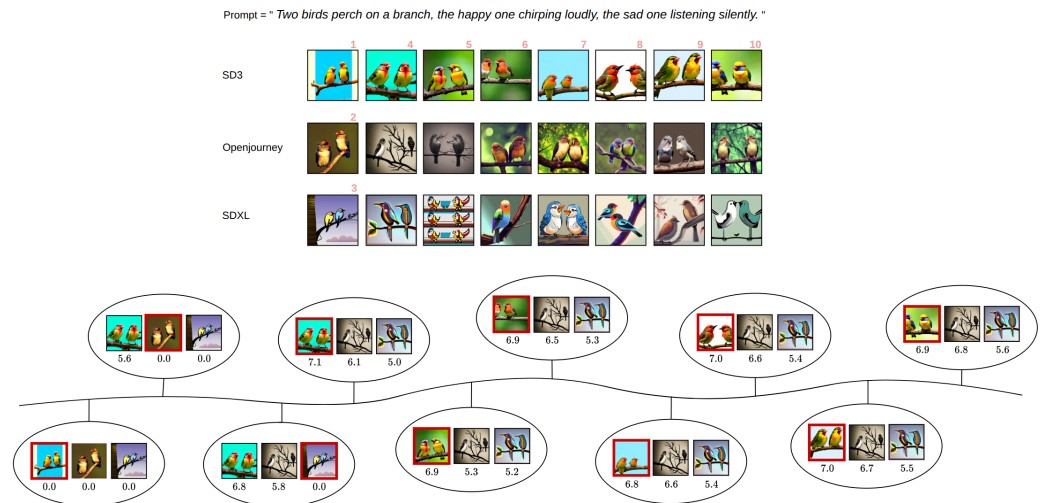

(a) Results with UCB on real data.

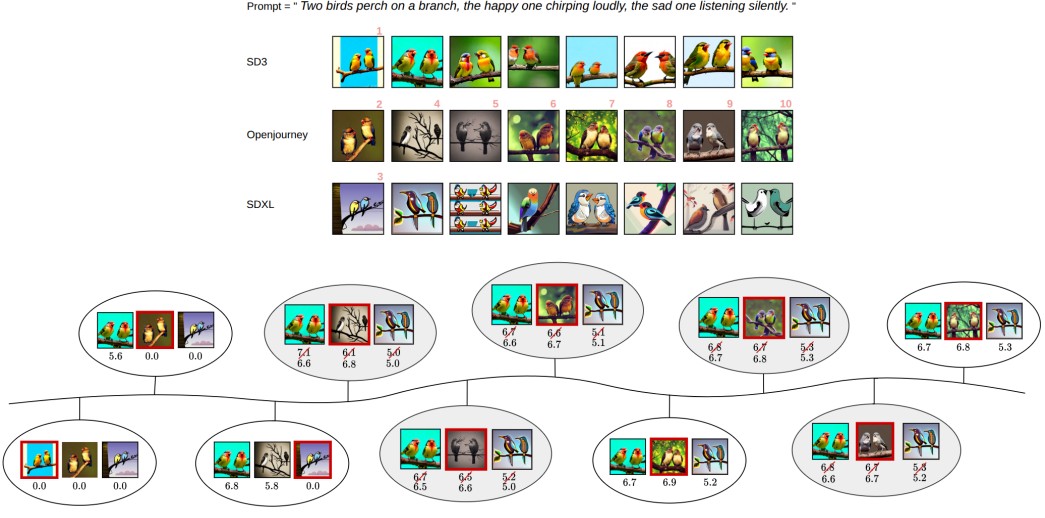

(b) Results with OSA on real data.

Figure 11: Comparison of methods on real data. (a) UCB results. (b) OSA results.

## K  ALGORITHMS

The detailed algorithms for the *Full Trajectory Attack* (Algorithm 1), *OSA Attack* (Algorithm 2), and *Trajectory-Free Attack* (Algorithm 3) are provided below. Each follows the same structure with three functions: `select_arm`, `find_perturbation`, and `update`, with differences noted in the comments.

---

**Algorithm 1:** Full Trajectory Attack

---

**Function** `select_arm`(*t, K, $\widetilde{A}$*)**:**

    **if** $t \leq K$ **then**

        **return** *t, False*

    **else**

        // Adversary selects an arm according to the target trajectory

        $arm \leftarrow \widetilde{A}_t$

        **return** *arm, True*

**Function** `find_perturbation`(*K*, S, G, F, $\mathcal{C}$, $\{\mu_i\}_{i=1}^K$, *use_NN*)**:**

    // Consider all the inequalities

    **for** $j \leftarrow 1$ **to** $K$, $j \neq arm$ **do**

        **if** *use_NN* **then**

            $d_j \leftarrow \mathrm{G}^{(arm)} - \mathrm{G}^{(j)}$

            $c_j \leftarrow \sqrt{\frac{2 \log t}{N_j}} - \sqrt{\frac{2 \log t}{N_{arm}}} + (\mathrm{F}^{(j)} - \mathrm{F}^{(arm)})$

        **else**

            $d_j \leftarrow \mathrm{S}^{(arm)} - \mathrm{S}^{(j)}$

            $c_j \leftarrow \sqrt{\frac{2 \log t}{N_j}} - \sqrt{\frac{2 \log t}{N_{arm}}} - \mu_j^\top d_j$

        Append $(d_j, c_j)$ to $\mathcal{C}$

    $\mathcal{M} \leftarrow \varnothing$

    **foreach** $(d, c) \in \mathcal{C}$ **do**

        $\mathcal{M} \leftarrow \mathcal{M} \cup \{\delta^\top d \geq c + 10^{-6}\}$

    $\delta \leftarrow$ solve QP in Eq. equation 2 with constraints $\mathcal{M}$

    **return** $\delta$

**Function** `update`(X, $\mathrm{S}^{(arm)}$, $\mathrm{F}^{(arm)}$, $\mathrm{G}^{(arm)}$, *use_NN*)**:**

    $N_{arm} \leftarrow N_{arm} + 1$

    **if** *use_NN* **then**

        $\mathrm{G}^{(arm)} \leftarrow \mathrm{G}^{(arm)} + \frac{1}{N_{arm}}(\nabla NN_\theta(\mathrm{X}) - \mathrm{G}^{(arm)})$

        $\mathrm{F}^{(arm)} \leftarrow \mathrm{F}^{(arm)} + \frac{1}{N_{arm}}(NN_\theta(\mathrm{X}) - \mathrm{F}^{(arm)})$

    **else**

        $\mathrm{S}^{(arm)} \leftarrow \mathrm{S}^{(arm)} + \frac{1}{N_{arm}}(\mathrm{X} - \mathrm{S}^{(arm)})$

// Inputs: $\{\mathcal{D}_i\}_{i=1}^K$ Logged data, $\widetilde{A}$ is target trajectory, $\{\mu_i\}_{i=1}^K$ are inferred means, use_NN is a boolean flag, $NN_\theta$ is the reward model

**Input:** $K, d, T, \{\mathcal{D}_i\}_{i=1}^K,$ use_NN, $NN_\theta, \{\mu_i\}_{i=1}^K, \widetilde{A}$

**Output:** Perturbation $\delta$

// $N$: counts, S: sample mean, G: mean gradient, F: mean network output, $\mathcal{C}$: constraint set

Initialize $N, \mathrm{S}, \mathrm{G}, \mathrm{F} \leftarrow 0$

Initialize $\mathcal{C} \leftarrow \varnothing, \delta \leftarrow 0$

**for** $t \leftarrow 1$ **to** $T$ **do**

    $arm, do\_attack \leftarrow$ `select_arm`(*t, K, $\widetilde{A}$*)

    **if** $do\_attack$ **then**

        $\delta \leftarrow$ `find_perturbation`(*K*, S, G, F, $\mathcal{C}$, $\{\mu_i\}_{i=1}^K$, *use_NN*)

    Pull $arm$, observe $\mathrm{X} \sim \mathcal{D}_{arm}$

    `update`(X, $\mathrm{S}^{(arm)}$, $\mathrm{F}^{(arm)}$, $\mathrm{G}^{(arm)}$, *use_NN*)

**return** $\delta$

---

---

**Algorithm 2:** OSA Attack: Online Score-Aware Attack

---

**Function** `select_arm`$(t, K, N, \mathrm{R})$**:**

  **if** $t \leq K$ **then**

    |  **return** $t$, *False*

  **else**

    // Select the runner-up arm for attack

    **for** $j \leftarrow 1$ **to** $K$ **do**

      |  $\mathrm{UCB}[j] \leftarrow \mathrm{R}^{(j)} + \sqrt{\frac{2\log t}{N_j}}$

    $arm \leftarrow \arg\max_j \mathrm{UCB}[j]$

    // Attack if the arm with the highest UCB is the optimal arm

    $do\_attack \leftarrow (arm == 1)$

    **return** $arm$, $do\_attack$

**Function** `find_perturbation`$(\mathrm{S}, \mathrm{G}, \mathrm{F}, \mathcal{C}, \mu_1, use\_NN)$**:**

  // Construct inequality constraints to prevent selection of optimal arm

  **if** *use_NN* **then**

    $d \leftarrow \mathrm{G}^{(arm)} - \mathrm{G}^{(1)}$

    $c \leftarrow \left(\sqrt{\frac{2\log t}{N_1}} - \sqrt{\frac{2\log t}{N_{arm}}}\right) + (\mathrm{F}^{(1)} - \mathrm{F}^{(arm)})$

  **else**

    $d \leftarrow \mathrm{S}^{(arm)} - \mathrm{S}^{(1)}$

    $c \leftarrow \left(\sqrt{\frac{2\log t}{N_1}} - \sqrt{\frac{2\log t}{N_{arm}}}\right) - \mu_1^{\top} d$

  Append $(d, c)$ to $\mathcal{C}$

  $\mathcal{M} \leftarrow \varnothing$

  **foreach** $(d, c) \in \mathcal{C}$ **do**

    |  $\mathcal{M} \leftarrow \mathcal{M} \cup \{\delta^{\top} d \geq c + 10^{-6}\}$

  $\delta \leftarrow$ solve QP in Eq. equation 2 with constraints $\mathcal{M}$

  **return** $\delta$

**Function** `update`$(\mathrm{X}, \mathrm{R}, \mathrm{S}^{(arm)}, \mathrm{F}^{(arm)}, \mathrm{G}^{(arm)}, use\_NN, \delta)$**:**

  $N_{arm} \leftarrow N_{arm} + 1$

  **if** *use_NN* **then**

    $\mathrm{G}^{(arm)} \leftarrow \mathrm{G}^{(arm)} + \frac{1}{N_{arm}}(\nabla NN_\theta(\mathrm{X}) - \mathrm{G}^{(arm)})$

    $\mathrm{F}^{(arm)} \leftarrow \mathrm{F}^{(arm)} + \frac{1}{N_{arm}}(NN_\theta(\mathrm{X}) - \mathrm{F}^{(arm)})$

  **else**

    |  $\mathrm{S}^{(arm)} \leftarrow \mathrm{S}^{(arm)} + \frac{1}{N_{arm}}(\mathrm{X} - \mathrm{S}^{(arm)})$

  // Update empirical rewards using current $\delta$

  Update $R$ using $\delta$

// Inputs: $\{\mathcal{D}_i\}_{i=1}^{K}$ Logged data, $\{\mu_i\}_{i=1}^{K}$ are inferred means, use_NN is a boolean flag, $NN_\theta$ is the reward model

**Input:** $K, d, T, \{\mathcal{D}_i\}_{i=1}^{K}$, `use_NN`, $NN_\theta$, $\{\mu_i\}_{i=1}^{K}$ are inferred means

**Output:** Perturbation $\delta$

// $N$: counts, S: sample mean, G: mean gradient, F: mean network output, R: empirical rewards, $\mathcal{C}$: constraint set

Initialize $N, \mathrm{S}, \mathrm{G}, \mathrm{F} \leftarrow 0$

Initialize $\mathcal{C} \leftarrow \varnothing, \delta \leftarrow 0$

**for** $t \leftarrow 1$ **to** $T$ **do**

  $arm, do\_attack \leftarrow$ `select_arm`$(t, K, N, \mathrm{R})$

  **if** $do\_attack$ **then**

    |  $\delta \leftarrow$ `find_perturbation`$(\mathrm{S}, \mathrm{G}, \mathrm{F}, \mathcal{C}, \mu_1, use\_NN)$

  Pull $arm$, observe $\mathrm{X} \sim \mathcal{D}_{arm}$

  `update`$(\mathrm{X}, \mathrm{R}, \mathrm{S}^{(arm)}, \mathrm{F}^{(arm)}, \mathrm{G}^{(arm)}, use\_NN, \delta)$

**return** $\delta$

---

---

**Algorithm 3:** Trajectory-Free Attack

---

**Function** `select_arm(t, K)`:

  **if** $t \leq K$ **then**
    | **return** *t, False*
  **else**
    // Select a sub-optimal arm in round-robin order
    $arm \leftarrow$ next sub-optimal arm
    **return** *arm, True*

**Function** `find_perturbation(S, G, F, C, μ₁, use_NN)`:

  // Construct constraints only w.r.t. optimal arm
  **if** *use_NN* **then**
    $d \leftarrow G^{(arm)} - G^{(1)}$
    $c \leftarrow \left(\sqrt{\frac{2\log t}{N_1}} - \sqrt{\frac{2\log t}{N_{arm}}}\right) + (F^{(1)} - F^{(arm)})$
  **else**
    $d \leftarrow S^{(arm)} - S^{(1)}$
    $c \leftarrow \left(\sqrt{\frac{2\log t}{N_1}} - \sqrt{\frac{2\log t}{N_{arm}}}\right) - \mu_1^\top d$
  Append $(d, c)$ to $\mathcal{C}$
  $\mathcal{M} \leftarrow \varnothing$
  **foreach** $(d, c) \in \mathcal{C}$ **do**
    | $\mathcal{M} \leftarrow \mathcal{M} \cup \{\delta^\top d \geq c + 10^{-6}\}$
  $\delta \leftarrow$ solve QP in Eq. equation 2 with constraints $\mathcal{M}$
  **return** $\delta$

**Function** `update(X, S^(arm), F^(arm), G^(arm), use_NN)`:

  $N_{arm} \leftarrow N_{arm} + 1$
  **if** *use_NN* **then**
    $G^{(arm)} \leftarrow G^{(arm)} + \frac{1}{N_{arm}}(\nabla NN_\theta(X) - G^{(arm)})$
    $F^{(arm)} \leftarrow F^{(arm)} + \frac{1}{N_{arm}}(NN_\theta(X) - F^{(arm)})$
  **else**
    | $S^{(arm)} \leftarrow S^{(arm)} + \frac{1}{N_{arm}}(X - S^{(arm)})$

// Inputs: $\{\mathcal{D}_i\}_{i=1}^K$ Logged data, $\{\mu_i\}_{i=1}^K$ are inferred means, use_NN is a boolean flag, $NN_\theta$ is the reward model

**Input:** $K, d, T, \{\mathcal{D}_i\}_{i=1}^K, \{\mu_i\}_{i=1}^K,$ `use_NN`

**Output:** Perturbation $\delta$

// $N$: counts, S: sample mean, G: mean gradient, F: mean network output, $\mathcal{C}$: constraints

Initialize $N, S, G, F \leftarrow 0$

Initialize $\mathcal{C} \leftarrow \varnothing, \delta \leftarrow 0$

**for** $t \leftarrow 1$ ***to*** $T$ **do**

  $arm, do\_attack \leftarrow$ `select_arm(t, K)`
  **if** *do_attack* **then**
    | $\delta \leftarrow$ `find_perturbation(S, G, F, μ₁, C, use_NN)`
  Pull $arm$, observe $X \sim \mathcal{D}_{arm}$
  `update(X, S^(arm), F^(arm), G^(arm), use_NN)`

**return** $\delta$

---

## L    THE USE OF LARGE LANGUAGE MODELS (LLMS)

We used Large Language Models (LLMs) only in a limited way, specifically for minor writing polish and phrasing suggestions.

