# OpenReview forum: "Efficient Adversarial Attacks on High-dimensional Offline Bandits"
_ICLR.cc/2026/Conference — ICLR 2026 Poster_

### Official Review · Reviewer_HY31 · 2025-10-28

**Soundness:** 3
**Presentation:** 3
**Contribution:** 3
**Rating:** 6
**Confidence:** 3

**Summary:**

This work represents a breakthrough contribution to the field of adversarial attacks on bandit algorithms, with profound theoretical, methodological, and practical significance. By systematically revealing the extreme vulnerability of offline multi-armed bandit algorithms to reward model perturbations in high-dimensional scenarios, the authors construct a rigorous and cohesive research framework integrating theory, algorithms, and experiments. The novel threat model, clear theoretical characterization of high-dimensional vulnerability, efficient attack strategies, and comprehensive validation on real-world reward models address a critical gap in the community—especially amid the growing reliance of generative model evaluation on bandit methods. However, the work faces non-trivial limitations in threat model assumptions, defense robustness, experimental depth, theoretical generalization, and ethical discussion.

**Strengths:**

Offline bandits have been widely adopted in generative model evaluation, but existing adversarial research mostly focuses on online settings or direct reward tampering. This paper pioneers the systematic study of a new threat model—"attackers only perturb pre-trained reward model weights"—filling a critical gap in the field. The problem definition is highly relevant to real-world evaluation scenarios, providing valuable insights for the community.

**Weaknesses:**

The paper assumes the attacker has full access to the same offline dataset as the victim, can modify the reward model weights before the victim trains the bandit, and the victim will use the tampered weights for evaluation. In practical evaluation scenarios, data and hyperparameters are often not fully disclosed to attackers, which limits the work’s practical applicability. The proposed data shuffling defense is only effective under the ideal setting where "the attacker fully knows the original sample order." If the attacker constructs perturbations based on statistical properties rather than sample order, the defense may fail. Additionally, there is no theoretical quantification of the utility-robustness tradeoff for the defense. Theorem 3.4 assumes the data distribution is a product measure with identity covariance matrix, and the mean of the optimal arm is orthogonal to that of the second-best arm. This idealized setting may not hold for high-dimensional natural images, limiting the theoretical results’ generalization.

**Questions:**

Extend the analysis to "partial information" (e.g., attacker has access to a subset of offline data) and "black-box reward model" scenarios, discussing the feasibility of attacks under these more realistic settings. Conduct sensitivity analysis on data sampling errors and model parameter uncertainty, quantifying how deviations from the ideal assumption affect attack performance. Relax the idealized assumptions in Theorem 3.4, extending the theoretical results to non-identity covariance matrices or correlated features.

---

> ### Author Response · Authors · 2025-11-19
>
> We would like to thank the reviewer for their careful reading of our work, positive feedback, and insightful comments. The reviewer has mentioned three concerns/questions, which we attempt to address as follows:
>
> - **Weakness 1**: "The paper assumes the attacker has full access to the same offline dataset as the victim, can modify the reward model weights before the victim trains the bandit, and the victim will use the tampered weights for evaluation. In practical evaluation scenarios, data and hyperparameters are often not fully disclosed to attackers, which limits the work’s practical applicability."
>
> **Our response**: The reviewer is correct. However, we believe our assumptions remain relevant for two reasons:
>
> - First, many offline datasets are public benchmarks that are reused by thousands of researchers worldwide (see several of our references [1, 2]) and often have stream-like structure: for example, video, speech, or medical records of patients receiving treatment over time (which, surprisingly, are hard to shuffle). Therefore, our assumption is not unrealistic.
>
> - Second, another (more indirect) motivation for our setting is the threat of overfitting. Researchers often test multiple reward functions (e.g., with varying hyperparameters) on the same offline dataset due to the high cost of generating fresh samples for each bandit run. In such cases, no true adversary exists, but the effect can be modeled similarly. Standard complexity measures used to study overfitting (e.g., Rademacher complexity) already rely on adversarial worst-case reasoning. Although this direction is beyond the scope of our current paper, our work can serve as motivation for such analyses.
>
> The reviewer’s suggestion regarding partially revealed data is indeed interesting and could be pursued in future research. We thank the reviewer for this insight and will include it as a potential future direction. However, conducting a thorough analysis of this aspect would likely be too extensive for the current manuscript.
>
> ----
>
> - **Weakness 2**: "The proposed data shuffling defense is only effective under the ideal setting where the attacker fully knows the original sample order. If the attacker constructs perturbations based on statistical properties rather than sample order, the defense may fail. Additionally, there is no theoretical quantification of the utility-robustness tradeoff for the defense."
>
> **Our response**: The reviewer is correct that a more sophisticated, order-invariant attack could nullify the simple defense we propose. It is also true that we do not provide a theoretical utility–robustness tradeoff for this defense. However, we emphasize that our main contribution is the attack, not the defense. The defense is included only as a very simple strategy that mitigates at least part of the attack, intended to highlight that defense is possible and to motivate future work on more robust methods. We believe developing defenses at the level suggested by the reviewer would fall outside the scope of the current paper.
>
> ---
>
> - **Weakness 3**: "Theorem 3.4 assumes the data distribution is a product measure with identity covariance matrix, and the mean of the optimal arm is orthogonal to that of the second-best arm. This idealized setting may not hold for high-dimensional natural images, limiting the theoretical results’ generalization."
>
> **Our response**: The reviewer is correct; however, none of these assumptions are essential or fundamental. Each of them can be removed, though doing so would lead to significantly more complicated (and less readable) formulations. We chose to keep the setting as minimal as possible while preserving the core ideas of the proof, avoiding lengthy expressions that reduce clarity and intuition.
>
> In the revised version, we generalize the theoretical results and remove both the identity-covariance assumption and the orthogonality requirement. In particular, we only need $\lambda_{\min}(\Sigma)$ to be strictly positive (meaning the covariance matrices must be positive definite with strictly bounded eigenvalues from below), and $\theta( \mu^{\star},\mu_2)$ the inner-product angle between the optimal mean vector $\mu^\star$ and the second-best mean vector $\mu_2$) to be strictly above zero.
>
> ---
>
> We hope the reviewer’s concerns have been adequately addressed. If so, we would appreciate it if they could consider raising their score.
>
> ----
>
> [1] Behnamnia et al., “Log-Sum-Exponential Estimator for Off-Policy Evaluation and Learning,” ICML, 2025.
>
> [2] Liu and Shroff, “Data poisoning attacks on stochastic bandits,” ICML, 2019.

---

> ### Author Response · Authors · 2025-11-26
>
> Dear Reviewer HY31,
>
> Thank you again for your thoughtful review of our work. As we approach the end of the author–reviewer discussion period, we wanted to check whether you have any remaining concerns or questions. We would be happy to address them or engage in further discussion.

---

### Official Review · Reviewer_8Gpp · 2025-11-02

**Soundness:** 3
**Presentation:** 3
**Contribution:** 2
**Rating:** 4
**Confidence:** 5

**Summary:**

This paper studied offline adversarial attacks against linear UCB algorithm in high-dimensional space. The attacker has knowledge of the offline dataset associated with each arm, and has the ability to perturb the reward function. There are two scenarios considered in this paper - linear reward function and neural network-based reward. The paper studied 3 different attack goals. Theoretical results are derived to show attack feasibility. Interestingly, the authors showed that when the problem dimension is higher, the attack is easier to achieve. Experiments are done to demonstrate the effectiveness of the proposed attacks.

**Strengths:**

The paper extended traditional attacks on multi-armed bandits to high-dimensional space, and studied the problem from a different perspective. Instead of focusing on analyzing the attack cost, this paper targets analyzing the behavior of attack as the problem dimension grows. This is a new angle and may bring interesting topics to the community.

Both theoretical analysis and empirical study are performed the analyze the behavior of the proposed attacks, as the dimension of the bandit grows. An interesting observation is that the attack is easier as the dimension increases.

**Weaknesses:**

The problem setup is hard to justify in the following sense.

1. The attacker can perturb the reward function. This is too strong power. Traditional attacks only require attackers to perturb instantiated rewards, rather than the underlying reward mechanism. This is saying the attacker need to be able to change the underlying environment completely, which is too demanding.

2. Even if the bandit algorithm is forced to follow certain behaviors under attack, the data observed in each time step is still drawn from an pre-specified offline dataset. This is very weird setup, because attacker should have changed the learner's behavior, and the data should no longer be clean and fresh. This is a critical caveat in the problem setup.

3. The attack feasibility is hard to satisfy. The dimension must upper bound KT, which is a very strict constraint.

**Questions:**

Please help clarify my 3 concerns in the weakness part.

---

> ### Author Response · Authors · 2025-11-19
>
> We would like to thank the reviewer for their insightful comments and feedback. The reviewer has raised 3 concerns/questions, to which we provide a point-to-point response below.
>
> - **Weakness 1**: "The attacker can perturb the reward function. This is too strong power. Traditional attacks only require attackers to perturb instantiated rewards, rather than the underlying reward mechanism. This is saying the attacker needs to be able to change the underlying environment completely, which is too demanding."
>
> **Our response**: Our setting, similar to most modern and practical MAB formulations, assumes that pulling an arm produces a data sample, not a direct reward value, and the bandit/learner then evaluates the reward based on that sample. For example, an image generator or an LLM returns an image or text when queried (equivalent to pulling an arm), and the learner assesses a reward (e.g., fidelity or quality) using some reward function. Such reward functions may be linear or neural networks. Here, sample generation is handled by the environment, to which we assume no access, whereas reward functions are typically derived from publicly available pretrained models that can be poisoned and later downloaded by an unknowing learner for bandit training. Several of the cited works, including [1, 2], in our Introduction highlight exactly this kind of setting.
>
> Another point to mention is that we believe our setting is more realistic in practice than previous attack strategies. Instantiated attack perturbation requires the adversary to be online, active, and present inside the private server where the bandit is trained. This is actually too much to assume from the adversary’s side. We (both theoretically and experimentally) show that poisoning the reward function **before** training even begins can still cause equally significant damage, while the attack itself is much simpler to perform. We believe it is important for the community to recognize that such a vulnerability exists in offline bandits, which may motivate future efforts toward developing defense mechanisms.
>
>
> In the revised version, we added this explanation to Appendix I to clarify our novel threat model.
>
> ---
>
> - **Weakness 2**: "Even if the bandit algorithm is forced to follow certain behaviors under attack, the data observed at each time step is still drawn from a pre-specified offline dataset. This is very weird, because the attacker should have changed the learner's behavior, and the data should no longer be clean and fresh. This is a critical caveat in the problem setup."
>
> **Our response**: This is precisely the standard offline bandit setup, where clean samples from the environment can be accessed in order whenever an arm is pulled. We make no assumptions beyond this typical setting. Importantly, we do not attack the samples themselves. They remain unchanged and clean. We only poison the reward function, which lies on the learner’s side. Note that altering the bandit’s behavior or trajectory does not alter the offline dataset or the samples’ ordering. It merely changes the order in which arms are pulled. When the bandit pulls arm $i$ for the $k$-th time, the same fixed sample is returned; only the time step at which this occurs differs due to the shifted trajectory.
>
> ----
>
> - **Weakness 3**: "The attack feasibility is hard to satisfy. The dimension must upper bound KT, which is a very strict constraint."
>
> **Our response**: Let us discuss two points. First, the condition $d \ge KT$ is required only for full-trajectory attacks. Our stronger OSA attack does not require such large dimensions to work provably. Even full-trajectory attacks often succeed with much smaller dimensions, though our bound is worst-case. Second, high-dimensional settings, such as images or text embeddings, are commonplace in today’s practical MABs, making $d \ge KT$ easily satisfied in real applications. In Section 4.3 and Figure 11, our experiments further demonstrate the practicality of our attacks on several real-world datasets.
>
> ----
>
> We hope these responses address the reviewer’s concerns. If so, we kindly ask the reviewer to reconsider their score.
>
> ----
>
> [1] Rezaei et al., “Be more diverse than the most diverse: Optimal mixtures of generative models via mixture-UCB bandit algorithms,” ICLR, 2025.
>
> [2] Hu et al., “An online learning approach to prompt-based selection of generative models and LLMs,” ICML, 2024.

---

> ### Author Response · Authors · 2025-11-26
>
> Dear Reviewer 8Gpp,
>
> Thank you again for your thoughtful review of our work. As we approach the end of the author–reviewer discussion period, we wanted to check whether you have any remaining concerns or questions. We would be happy to address them or engage in further discussion.

---

### Official Review · Reviewer_sGeV · 2025-11-03

**Soundness:** 2
**Presentation:** 1
**Contribution:** 1
**Rating:** 2
**Confidence:** 3

**Summary:**

The paper studies reward poisoning for stochastic multi-armed bandits under two reward function classes. It proposes three attack strategies and derives bounds on the maximum per-round corruption required to force errors against upper-confidence-bound style algorithms. The empirical section illustrates the practical impact of these attacks.

**Strengths:**

1. The problem setting is stated clearly, and the three corruption procedures are precisely specified.

**Weaknesses:**

1. The related work on corruption robust multi-armed bandits is not discussed at all, only a few mentioned in Appendix. I suggest the authors to include important works such as [1, 2, 3] and directly compare them by total amount of corruption ( $\mathbb{E}[ \sum_t  \delta^\top X_t] $).

2. The proposed algorithms is mostly examined against vanilla UCB, and briefly on greedy algorithms, both of which are not designed to be adversarially robust. What would be the minimum $\delta$ required against such robust MAB algorithms (e.g. [1, 2])?

3. Figure 1 is unclear. The x-axis uses “attack time (seconds)” while the rest of the paper uses discrete rounds. Either convert everything to rounds or explain the mapping between wall-clock time and rounds.

[1] Lykouris, T., Mirrokni, V. and Paes Leme, R., 2018. "Stochastic bandits robust to adversarial corruptions". In Proceedings of the 50th Annual ACM SIGACT Symposium on Theory of Computing.
[2] Gupta, A., Koren, T. and Talwar, K., 2019. "Better algorithms for stochastic bandits with adversarial corruptions". In Conference on Learning Theory.
[3] Hajiesmaili, M., Talebi, M.S., Lui, J. and Wong, W.S., 2020. "Adversarial bandits with corruptions: Regret lower bound and no-regret algorithm". Advances in Neural Information Processing Systems.

**Questions:**

1. How does the proposed poisoning perform against robust variants of UCB such as [1]?
2. Theorem 3.4 appears to imply that as $d \rightarrow \infty$, the amount corruption needed goes to zero which is contradictory as at least $\Delta_{min}$(reward gap between best and second best arm) per round corruption is needed to induce suboptimal play for $T \ge \frac{K}{\Delta_{min}^2}$. Could you elaborate on how $d$, $\Delta_{min}$, and $T$ are scaled in the theorem to avoid this apparent discrepancy?

[1] Niss, L. &amp; Tewari, A.. (2020).  What You See May Not Be What You Get: UCB Bandit Algorithms Robust to $\varepsilon$-Contamination. Proceedings of the 36th Conference on Uncertainty in Artificial Intelligence.

---

> ### Author Response · Authors · 2025-11-19
>
> We thank the reviewer for their insightful comments. Below, we provide a point-by-point response. We begin with Weakness 1.
>
> - **Weakness 1**: "The related work on corruption-robust MABs is not discussed thoroughly; I suggest including works such as [1,2,3] and directly comparing them by the total amount of corruption  ($\mathbb{E}[\sum_t \delta^\top X_t]$)."
>
> **Response to Weakness 1**: We appreciate the suggestion and have added the recommended citations [1,2,3], along with a dedicated section 4.8 to discuss and compare our framework with corruption-based attacks. We have specifically conducted experiments to compare Lykouris et al. (2018), and  Niss and Tewari (2020). Please refer to the revised manuscript, and also the next parts of this official comment.
>
> That said, it is important to clarify a key conceptual difference between our setting and the reward-corruption attacks referenced by the reviewer. Existing corruption-robust MAB works assume an adversary who actively modifies the realized rewards during the bandit interaction, i.e., the adversary is active, present throughout training, and directly alters the reward feedback as it is generated. In contrast, our attack perturbs the internal weights of the reward model **before** bandit training begins. The adversary does **not** have access to the environment where the bandit is run, which we believe is a much more realistic setting. This pre-trained, offline threat model, where a victim downloads a compromised reward model from a repository such as HuggingFace, has not been studied in prior MAB attack literature.
>
> For example, consider selecting the best among $K$ generative models using a downloaded reward function (e.g., a pretrained neural network to assess the fidelity or diversity of the generated images) and a standard bandit algorithm such as UCB. We show that a tiny, imperceptible change to the reward model’s weights can fully hijack the bandit’s behavior. An attacker only needs to upload this subtly poisoned model; the victim unknowingly downloads and uses it, without any adversarial presence in the private server where training occurs. This type of “planted” perturbation is especially (and provably) potent in high-dimensional settings such as images and remains effective without any online interaction, unlike many corruption-based attacks that require continuous adversarial access.
>
> For these reasons, comparing attack budgets across these settings is, in our view, not directly meaningful: the underlying threat models differ substantially. Please also refer to the comments from Reviewer HY31. Nevertheless, we have incorporated the suggested works into the related-work section and thank the reviewer again for pointing them out.
>
> ---
>
> Weakness 2 and Question 1 ask whether robust variants of UCB can defend against our proposed attack.
>
> - **Weakness 2**: "The proposed method is mostly examined against vanilla UCB, and briefly on greedy algorithms, both of which are not designed to be adversarially robust. What happens when tested with robust bandit algorithms (e.g., [1,2])?"
>
> - **Question 1**: "How does the proposed poisoning perform against robust variants of UCB such as [1]?"
>
> **Response to Weakness 2**: In the main paper, for clarity and consistency with our theoretical development, we focused on widely used bandit algorithms such as ETC and UCB. These are standard in practice, and our results show that even these commonly deployed algorithms are highly vulnerable to our attack.
>
> Following the reviewer’s suggestion, we additionally evaluated our attack during the rebuttal phase against the Fast–Slow algorithm of Lykouris et al., a representative robust bandit method. As shown in Figure 5 of the revised manuscript, this algorithm is also vulnerable. With a corruption parameter of $0.5$ (a typical setting in their own experiments), our attack succeeds 100% of the time, while keeping the perturbation norm small (around $0.3$). Moreover, as the corruption budget increases, which benefits the adversary, the required $\ell_2$-norm decreases, making the attack even easier to execute. We conducted these experiments for $T = 1000$, $d = 1000$, and $K \in \\{3,5\\}$. These results demonstrate that our poisoning method remains highly effective even against robust UCB variants.
>
> **Response to Question 1**: We evaluate our attack against the $\varepsilon$-contamination algorithm [Niss and Tewari (2020)], a robust variant of UCB. Figure 6 illustrates that for a relatively small contamination level ($\varepsilon = 0.15$), our attack achieves a $100\\%$ ASR while requiring only a small $\ell_2$-norm perturbation (approximately $0.2$). For this experiment, we use $T \in \\{100, 1000\\}$, $K \in \\{3, 5\\}$, $d = 1000$, $\alpha = 0.1$, $\sigma = 1$, and the $\alpha$-trimmed strategy within the $\varepsilon$-contamination algorithm.

---

> > ### Author Response · Authors · 2025-11-19
> >
> > - **Question 2**: "How is it possible to steer the bandit toward a false trajectory when the $\ell_2$-norm of the attack vector $\delta$ provably converges to zero as $d \to \infty$? It should be at least on the order of the gap between the best and second-best arm."
> >
> > **Response to Question 2**: The confusion arises from interpreting $\delta$ as a per-round reward perturbation vector. In our setting, $\delta \in \mathbb{R}^d$ is not a vector of reward corruptions across time. Instead, the reward perturbation at round $t$ is the *inner product* $\delta^\top X_t$, where both $\delta$ and $X_t$ are $d$-dimensional.
> >
> > Even when $\Vert\delta\Vert_2 \to 0$ as $d \to \infty$, the inner product $\delta^\top X_t$ can remain $O(1)$. For example, suppose each coordinate $\delta_i = O(1/d)$. For typical feature vectors $X_t$ whose coordinates are $O(1)$, we have
> > $\delta^\top X_t = \sum_{i=1}^d X_{t,i}\delta_i = O(1)$.
> > However, the norm of $\delta$ is
> > $\Vert\delta\Vert_2 = O([\sum_{i=1}^d (1/d^2)]^{1/2}) = O(d^{-1/2}) \to 0$.
> >
> > Thus, there is no contradiction: a vanishing $\ell_2$-norm does not prevent the attacker from inducing a constant-order reward shift through the inner product $\delta^\top X_t$.
> >
> > ----
> >
> > **Weakness 3**: “Figure 1 is unclear. The x-axis uses ‘attack time (seconds)’ while the rest of the paper uses discrete rounds. Either convert everything to rounds or explain the mapping between wall-clock time and rounds.”
> >
> > **Our response to Weakness 3**:
> > We believe the reviewer is referring to the **y-axis** of Figure 1, which indeed reports the attack time in seconds. In other parts of the paper, particularly in the theoretical and experimental discussion of the OSA attack, we refer to “rounds” or “time-steps.” These represent a fundamentally different concept.
> >
> > - **Rounds (or time-steps)** correspond to the iteration index $t = 1, \ldots, T$, which specifies the target points used in designing the perturbation vector $\delta$. In other words, the number of rounds determines the number of constraints in our quadratic optimization program. One key insight of our analysis and experiments is that achieving a 100% attack success rate does not require using all $T$ time-steps; in practice, a carefully chosen subset, sometimes only $O(\log T)$, suffices to generate an effective perturbation.
> >
> > - **Attack time (measured in seconds)**, on the other hand, indicates the total computation time required on our reference laptop to generate the attack. Figure 1 reports this quantity to demonstrate that OSA attacks can be computed very efficiently in practice.
> >
> > We will clarify this distinction in the revised version of the paper to avoid confusion.
> >
> > ----
> >
> > Once again, we sincerely thank the reviewer for their time and constructive feedback. We hope that our responses and the corresponding revisions to the manuscript address the concerns raised. We would be grateful if the reviewer would consider updating their score accordingly.

---

> ### Author Response · Authors · 2025-11-26
>
> Dear Reviewer sGeV,
> Thank you again for your thoughtful review of our work. As we approach the end of the author–reviewer discussion period, we wanted to check whether you have any remaining concerns or questions. We would be happy to address them or engage in further discussion.

---

### Author Response · Authors · 2025-11-19
**Global Response**

Dear Reviewers and AC,

We would like to sincerely thank you for carefully reading our work and providing valuable feedback. We have thoroughly reviewed and discussed all comments, and have prepared the following:

- We have provided point-by-point responses to every question and concern raised by each reviewer individually.

- We have submitted a revised manuscript. This version includes several new experiments and comparisons, as well as additional citations (many of which were requested by Reviewer sGeV).

We are continuing to expand our experimental results and refine the theoretical analysis. We will be happy to respond to any further questions or requests.

---

### Meta-Review · Area_Chair_PQLH · 2026-01-06

**Summary:**

Reviewer concerns were mainly:
- Threat model: edit-access to the reward model (8Gpp), read-access to the victim offline evaluation dataset (HY31), and whether a (simpler?) defense would also consider updating the dataset in response (8Gpp)
- Theoretical arguments: strong assumptions for an ideal scenario - identity cov matrices + orthogonal means between arms (HY31), vanishing perturbation as $d \to \infty$ (sGeV), requiring $KT < d$ (8Gpp)
- Missing critical comparisons to robust MAB, confined to a literature review in the appendix, e.g. Lykouris et al., 2018 (sGeV)
- Issues with the evaluation (Section 4.8): proposed defense relies on sample ordering and how shuffling can help the attack (HY31)

**Reviewer Concerns:**

The authors clarify a number of critical points:
- Validate the threat model noting increasing reliance on pre-trained reward models served to victim users, and by the same token relying on open-source evaluation datasets as one standard practice
- Clarify some of the assumptions needed for the theory: number of arms $T$ and generative models $K$ is still much lower than (latent) image dimension $d$ justifying $KT < d$.  The exponential dependence on the dimension also clearly justifies vanishing perturbations.  The experimental results further validate this assumption in realistic settings. (Also note the distinction to full-trajectory attacks.)
- Comparison to robust MAB (requested by sGeV) - noting conceptual differences (this paper: offline, existing work: online).
- Authors acknowledge the lack of theoretical utility–robustness tradeoff for this defense, noting that the main contribution is the attack not the defense.

Additional comments:
- In the very first paragraph, I found it less useful to cite key papers of the relevant application domains (eg. diffusion models) rather than papers that actually apply bandits to those domains.  From one perspective, readers are much more likely looking for the latter rather than the former.
- Nit @ L062: new paragraph

**Reviewer Scores:**

Although initial ratings came as 6/4/2, there is significant misunderstanding on the reviewer's part which may have (understandably) shifted their first impression a bit prematurely.  While it may not be feasible to predict how our particular reviewers may have changed their positions, I'd expect a final score closer to 6.

Overall, the paper introduces an interesting and relevant attack relevant to the evaluation of generative models as already adopted by the community, supported by valuable theory and substantial experimental evidence -- codes already available.  This makes for a decent paper to ICLR this year.

---

### Decision · Program_Chairs · 2026-01-26

Accept (Poster)